# Memory Retrieval and Consolidation in Large Language Models through Function Tokens

## Abstract

The remarkable success of large language models (LLMs) stems from their ability to consolidate vast amounts of knowledge into the memory during pre-training and to retrieve it from the memory during inference, enabling advanced capabilities such as knowledge memorization, instruction-following and reasoning. However, the mechanisms of memory retrieval and consolidation in LLMs remain poorly understood. In this paper, we propose the function token hypothesis to explain the workings of LLMs: During inference, function tokens activate the most predictive features from context and govern next token prediction (memory retrieval). During pre-training, predicting the next tokens (usually content tokens) that follow function tokens increases the number of learned features of LLMs and updates the model parameters (memory consolidation). Function tokens here roughly correspond to function words in linguistics, including punctuation marks, articles, prepositions, and conjunctions, in contrast to content tokens. We provide extensive experimental evidence supporting this hypothesis. Using bipartite graph analysis, we show that a small number of function tokens activate the majority of features. Case studies further reveal how function tokens activate the most predictive features from context to direct next token prediction. We also find that during pre-training, the training loss is dominated by predicting the next content tokens following function tokens, which forces the function tokens to select the most predictive features from context.

## 1 Introduction

Large Language Models (LLMs) (OpenAI, 2022; Brown et al., 2020; OpenAI, 2023; Anthropic, 2024; Team et al., 2023; Liu et al., 2024) have demonstrated remarkable capabilities. They possess strong knowledge memorization abilities, ranging from remembering simple factual knowledge (e.g., *The capital of the United States is Washington, D.C.*) to the verbatim reproduction of lengthy passages (e.g., *Recite Martin Luther King Jr's "I Have a Dream" speech word by word*). Beyond that, LLMs also exhibit strong general skills, such as instruction following (Ouyang et al., 2022; Wei et al., 2021) (e.g., *As a financial analyst: explain quantitative tightening, then list three stock market impacts.*) and reasoning (Wei et al., 2022; Kojima et al., 2022) (e.g., *The streets are wet and the sidewalks are slick. What is the most likely explanation?*).

In the human brain, long-term memory forms through synaptic consolidation, where the synapses between neurons are strengthened, ultimately creating neural circuits that store knowledge (Josselyn & Tonegawa, 2020). Inspired by this biological mechanism, artificial neural networks have been developed. These systems consist of neurons linked by weighted connections, and their weights (parameters) are obtained by training on data. The weights of a neuron determines how it responds to its inputs to produce an activation (Geva et al., 2021). A technique utilizing Sparse Autoencoders (SAEs) (Cunningham et al., 2023) has been developed recently to analyze Transformer-based LLMs (Vaswani et al., 2017). It enables the decomposition of neuron activations into interpretable features, providing insights into how the circuits within the Transformer's layers are composed of these interpretable features (Elhage et al., 2022; Chen et al., 2025b; Hendel et al., 2023).

Despite significant progress in understanding LLM neuron activations, the memory mechanisms remain poorly understood. In particular, two fundamental questions are still not well addressed: (1) How is the memory retrieved during inference? and (2) How is the memory consolidated during pre-training? In this paper, we present our investigation into these questions. We find that analyzing

Figure 1: Function tokens can dynamically activate the most predictive features from the context to guide the next-token prediction. For example, the function token 'in' reactivates features 'J.K. Rowling' and 'Location' from context (while suppressing feature 'French') and activates 'England' to predict 'Britain'. In contrast, the content token 'Harry' activates feature 'Harry Potter'.

from the perspective of function tokens and content tokens can help unravel the mystery of memory retrieval and memory consolidation.

In linguistics, function words are words that have little semantic meanings but play crucial grammatical and connective roles within and between sentences, such as articles, prepositions, and conjunctions (Carnap, 2014). In contrast, content words are words that convey semantically explicit and rich meanings. The distribution of words in natural language follows Zipf's law (Kanwal et al., 2017). In this distribution, function words occur with disproportionately high frequencies, occupying the head, while content words appear with much lower frequencies, forming the long tail. LLMs utilize tokens, which may represent words, sub-words, or punctuation marks. In our work, for ease of experimentation, we automatically classify tokens into 'function tokens' and 'content tokens' based on their frequencies in the pre-training corpus, using this as an approximation of the linguistic concepts.

To investigate the role of function tokens during inference, we construct a bipartite graph connecting tokens to features obtained via SAE decomposition. We show that, although few in number, function tokens activate a large proportion of the LLM's features. Furthermore, our case studies show that the activation patterns for function tokens differ from those for content tokens. Function tokens dynamically reactivate predictive features from the context, whereas content tokens show little evidence of this effect. To understand why feature activations are centered on function tokens, we conduct pre-training experiments. We track next-token prediction loss across four categories based on whether the current token and the next token are function or content tokens. We find that LLMs first learn to predict function tokens before gradually learning to predict content tokens, a process accompanied by an increase in the number of features and the learning of the parameters. Furthermore, pre-training is dominated by the prediction of content tokens that follow function tokens. These observations reveal why function tokens can access a large portion of the LLM's features. Based on these findings, we propose the Function Token Hypothesis (see an example in Figure 1).

In this paper, the LLMs are GPT-type models with a Transformer decoder architecture, obtained through pre-training and post-training (including SFT and RL) (Nakano et al., 2021; Radford et al., 2018). Both pre-training and inference are conducted autoregressively via next-token prediction. At each layer of the Transformer, a vector of activations (after the add-norm operation of FFN) can be created. SAE can be performed on this activation vector to obtain a linear combination of features. Here, knowledge refers to the LLM's parameters as well as all possible features that can be derived from them. Memory is the virtual system that stores the knowledge. Memory retrieval means the activations of features and circuits (Olah et al., 2020; Elhage et al., 2021; Wang et al., 2023; Merullo et al., 2024), while memory consolidation means the learning of the parameters to form and expand features and circuits.

**Function Token Hypothesis.** During inference, *function tokens activate the most predictive features from the context to direct the next-token prediction* (memory retrieval). During pre-training, *predicting content tokens based on the function tokens* drives the LLM to update its parameters to learn and expand features (memory consolidation).

The function token hypothesis is also supported by many phenomena observed in LLM research. For example, activations with unusually large magnitudes often occur at the initial tokens, periods, or newlines (Sun et al., 2024). Meaningless separator tokens disproportionately affect attention compared to semantically rich tokens (Chen et al., 2025a). The use of 'pivot tokens' during post-training can significantly enhance performance in response (Abdin et al., 2024). Training that concentrates

on high-entropy tokens also yields better performance (Wang et al., 2025a). We argue that these tokens are all function tokens that behave as the hypothesis predicts.

We believe that unraveling the important role of function tokens in LLM memory mechanisms not only enhances research on LLM interpretability but also provides insights for designing advanced learning algorithms, particularly those for enhancing alignment with human values.

The main contributions of this paper are summarized as follows:

- We demonstrate that during inference, function tokens are responsible for activating the most predictive features from the context to govern next-token prediction.
- We show that feature growth during pre-training is driven by the prediction of content tokens that follow function tokens.
- We propose the Function Token Hypothesis for explaining LLM memory mechanisms.

## 2 PRELIMINARY

### 2.1 MODEL MEMORY AND SUPERPOSITION PHENOMENON

**Feed-Forward Network as Key-Value Memory** Existing work views the Feed-Forward Network (FFN) layer in each block of a Transformer as a key-value memory or a neural memory (Geva et al., 2021). Specifically, the FFN can be formulated as (bias terms are omitted, as in common practice):

$$\mathbf{z} = \mathrm{ReLU}(\mathbf{x} \cdot \mathbf{W}_k^\top), \quad \mathbf{y} = \mathbf{z} \cdot \mathbf{W}_v, \tag{1}$$

Here, $\mathbf{x} \in \mathbb{R}^d$ is the input vector, $\mathbf{y} \in \mathbb{R}^d$ is the output vector, $\mathbf{z} \in \mathbb{R}^{d_m}$ is the weight vector, $\mathbf{W}_k \in \mathbb{R}^{d_m \times d}$ is the key matrix, $\mathbf{W}_v \in \mathbb{R}^{d_m \times d}$ is the value matrix, and $d_m$ is the memory size. The output vector $\mathbf{y}$ is the activation of the FFN layer, where each dimension corresponds to a neuron.

In the key-value memory interpretation, there are $d_m$ pairs of key vector and value vector. Each row of $\mathbf{W}_k \in \mathbb{R}^{d_m \times d}$ corresponds to a key vector $\mathbf{k}_i \in \mathbb{R}^d$ and each row of $\mathbf{W}_v \in \mathbb{R}^{d_m \times d}$ corresponds to a value vector $\mathbf{v}_i \in \mathbb{R}^d$. Given the input vector $\mathbf{x}$, the similarity between $\mathbf{x}$ and each of the key vectors $\mathbf{k}_i$ is first calculated as $z_i = \mathrm{ReLU}(\mathbf{x} \cdot \mathbf{k}_i^\top) \geq 0$, where ReLU acts as an unnormalized weighting function; the weighted sum of the corresponding value vectors based on the similarities is then calculated and output as $\mathbf{y} = \sum_{i=1}^{d_m} z_i \mathbf{v}_i$. The interpretation suggests that knowledge of the Transformer is represented in the parameters of the FFN layers. Note that Transformer attention layers also form key-value memories using softmax weighting.

**Superposition Phenomenon** Recent work on LLM interpretability shows the phenomenon of superposition (Elhage et al., 2022), in which features can be extracted from the activations of neurons in a Transformer-based LLM. The number of extracted features usually far exceeds the number of neurons. There exist many polysemantic neurons, each of which represents multiple meanings.

Through sparse dictionary learning, the activations of polysemantic neurons can be decomposed into monosemantic features, each corresponding to a distinct, human-interpretable concept, such as the Golden Gate Bridge (Templeton et al., 2024). A widely used method for dictionary learning is the Sparse Autoencoder (SAE), which learns to linearly decompose neuron activations through a reconstruction task. SAE decomposes an activation $\mathbf{y} \in \mathbb{R}^d$, typically the output of a specific layer, into a linear combination of features:

$$\mathbf{y} = \sum_{i=1}^n c_i \mathbf{f}_i = c_1 \mathbf{f}_1 + c_2 \mathbf{f}_2 + ... + c_n \mathbf{f}_n. \tag{2}$$

Here, $c_i$ denotes the coefficient for feature $\mathbf{f}_i \in \mathbb{R}^d$, the set of features $\{\mathbf{f}_i\}$ constitutes an approximately orthogonal basis, and $n$ represents the number of features, typically $n \gg d$.

Furthermore, the behavior of the LLM during generation can be partially controlled by steering the activations of features. For example, steering can control both specific concepts (e.g., Golden Gate Bridge (Templeton et al., 2024)) and behavioral patterns (e.g., sycophantic behavior (Panickssery et al., 2023)). Similar feature activation phenomena are observed in human memory recall, with empirical evidence supporting the existence of neurons representing either specific or general concepts.

### 2.2 FUNCTION TOKENS AND CONTENT TOKENS

We tokenized the SlimPajama-627B corpus (Soboleva et al., 2023), a widely used pre-training dataset, using the LLaMA-3.1 tokenizer and sampled 1 billion tokens for statistical analysis. We group the tokens into function tokens and content tokens based on their frequency. This leverages the linguistic fact that function words

| Token rank | Token text | Token Fraction | Cumulative Fraction | Document Coverage |
|---|---|---|---|---|
| 1 | , | 3.60% | 3.60% | 95.00% |
| 2 | the | 3.19% | 6.79% | 90.92% |
| 3 | . | 2.23% | 9.10% | 95.80% |
| 4 | and | 1.81% | 10.91% | 89.69% |
| 5 | of | 1.80% | 12.71% | 87.59% |
| 6 | to | 1.68% | 14.40% | 88.71% |
| 7 | white space | 1.59% | 15.99% | 81.35% |
| 8 | a | 1.33% | 17.32% | 87.62% |
| 9 | in | 1.16% | 18.48% | 86.04% |
| 10 | .\n | 0.90% | 19.39% | 84.58% |

(a) Zip'f distribution.  (b) The 10 most frequent tokens.

Figure 2: Token frequency statistics in SlimPajama-627B.

typically have higher frequency, while content words have lower frequency. Starting from the most frequent, we add tokens until the set covered 40% of all token occurrences, yielding 122 tokens labeled as function tokens; the rest are taken as content tokens. The resulting set of function tokens roughly corresponds to the function words defined in linguistics, with several exceptions like punctuation marks. The full list of function tokens appears in Appendix F.

**Token Frequency and Zipf's Law** As shown in Figure 2a, token frequency follows the Zipf's law (Piantadosi, 2014): $f(r) \propto r^{-\alpha}$, where $f(r)$ is the frequency of the token ranked $r$, revealing a fundamental property of natural language: a few tokens are used frequently, while most are used infrequently. For example, the 10 most frequent tokens account for 19.39% of the corpus (Figure 2b).

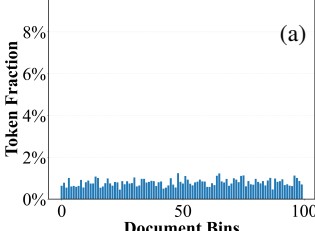 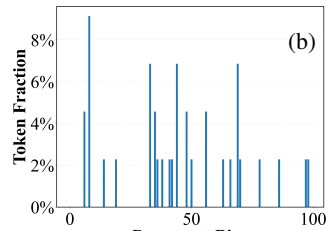 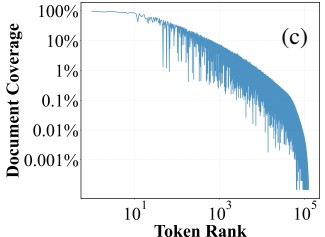

Figure 3: Distribution of function and content tokens. Document bins represent equal partitions of corpus. (a) Function token 'of' shows uniform, dense coverage across documents. (b) Content token 'Tokyo' shows sparse coverage. (c) Document coverage versus token rank follows a power-law.

**Document Coverage** A pre-training corpus contains a vast number of documents. High-frequency tokens are distributed uniformly across documents, while low-frequency tokens appear frequently within a limited number of documents (Rychlý, 2011), showing bursty distributions. For instance, as shown in Figure 3, the function token 'of' appears with similar frequency across documents, whereas the content token 'Tokyo' occurs only in a few. Thus, high-frequency tokens are utilized in nearly all training examples, while low-frequency tokens are used only in a small fraction of them.

As shown in Figure 3, token frequency strongly correlates with document coverage. Figure 2b lists the 10 most frequent tokens along with their document coverage. Formally, the document coverage of token $t$ is $\frac{|\{d \in D : t \in d\}|}{|D|}$, where $D$ denotes the entire set of documents.

## 3 MEMORY RETRIEVAL THROUGH FUNCTION TOKENS

We study the relationships between tokens and model features during inference. The results show that a small set of function tokens can activate most features. Our case study reveals how the same function tokens create different activation patterns in different contexts, leading to different outputs.

### 3.1 A FEW FUNCTION TOKENS ACTIVATE MOST FEATURES

We use Gemma2-9B (Team et al., 2024) for our analysis, as it provides both models of different sizes and open-source SAEs (Lieberum et al., 2024). Gemma Scope has SAEs with varying dictionary widths. Among these, we select the SAE with the largest dictionary width, $2^{20}$, to facilitate a more comprehensive feature decomposition.

To study how features are activated during inference, as illustrated in Figure 4, we construct a token-feature bipartite graph through the following steps: (1) Extract activations. We feed 10,000 randomly sampled raw documents from the SlimPajama validation dataset into Gemma2-9B, with

approximately 5 million tokens, and extract activations from the residual stream. We focus on three representative layers: layer 9 (shallow), layer 20 (middle), and layer 31 (deep). (2) Decompose features. For each layer, we apply the corresponding SAE to decompose token activations into sparse features. (3) Build

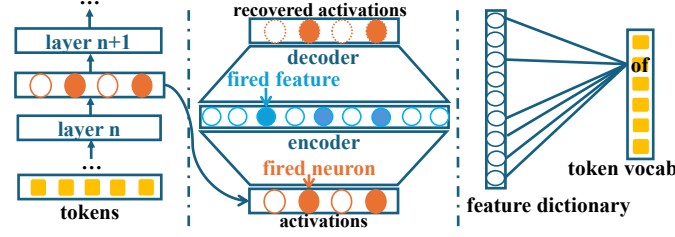

Figure 4: Construction of the bipartite graph.

bipartite graph. A token is linked to a feature if it activates the feature in a context. Each token-feature pair is connected by at most one edge, regardless of how many times the activation occurs.

The bipartite graph comprises two node types: tokens and features. The number of token nodes equals the vocabulary size, while the number of feature nodes (each connected to at least one token) is 965,635, 947,341, and 919,220 for the three layers, respectively. With a dictionary width of $2^{20}$, this yields activation rates of 92.1%, 90.3% and 87.7%, confirming sufficient coverage for analysis.

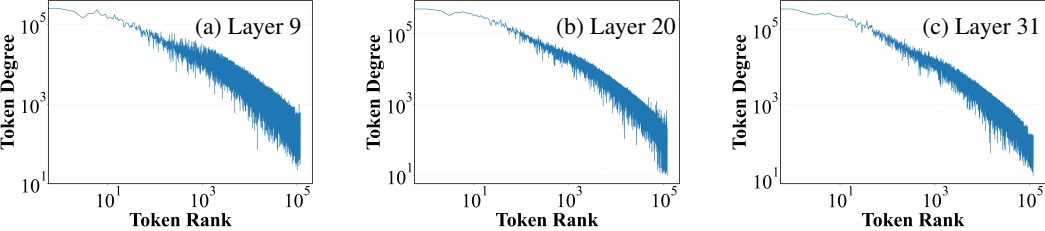

Figure 5: Token degrees in the bipartite graph, token ranked by frequency from the sampled data.

Figure 5 presents the degree of each token in the token-feature bipartite graph. The results reveal that a small set of function tokens can activate most features. Table 1 shows that the top 10 frequent tokens alone account for a substantial proportion of feature activations. In particular, in the middle layer, known to be the most expressive and interpretable (Panickssery et al., 2024; Soligo et al., 2025; Chen et al., 2025b), these tokens can activate more than 70% of the features, demonstrating function tokens' universal access to the feature space.

Table 1: Cumulative feature coverage by top-10 frequent tokens across different layers

| | Token | Cumulative Feature Coverage | | |
|---|---|---|---|---|
| Rank | Text | Layer 9 | Layer 20 | Layer 31 |
| 1 | . | 23.19% | 51.32% | 37.21% |
| 2 | , | 32.01% | 62.45% | 49.78% |
| 3 | the | 36.88% | 66.93% | 55.15% |
| 4 | \n | 39.68% | 71.30% | 59.86% |
| 5 | and | 41.21% | 71.97 % | 61.48% |
| 6 | to | 43.16% | 73.07 % | 63.30% |
| 7 | of | 46.00% | 74.43 % | 65.16% |
| 8 | white space | 47.44% | 75.70 % | 67.08% |
| 9 | a | 47.96% | 76.12% | 67.74% |
| 10 | in | 48.52% | 76.46% | 68.27% |

## 3.2 FEATURE REACTIVATION VIA FUNCTION TOKENS

Why can a small number of function tokens activate most features? We hypothesize that function tokens can reactivate the most predictive features, based on preceding contexts.

We design an experiment to examine this hypothesis. First, we identify three interpretable features in Gemma2-9B-it (Team, 2024): Feature 15261 corresponds to 'Speak Chinese', Feature 9591 corresponds to 'Russia', and Feature 13751 corresponds to 'UK'. The approach for identifying interpretable features is described in Appendix B. We then examine their activations during inference. We employ the following prompt template, wrapping the chat template used in Gemma2-9B-it.

We evaluate two prompts and record each token's feature activations. **Prompt 1**: *Answer the question in Chinese: What is the capital of Russia?* **Prompt 2**: *Answer the question in Chinese: What is the capital of UK?*

> **Prompt Template**
> <bos><start_of_turn>user
> {prompt} Directly answer the question<end_of_turn>
> <start_of_turn>model

As shown in Figure 6, for Prompt 1, the 'Speak Chinese' feature is first activated by the token 'Chinese', and the 'Russia' feature by the token 'Russia'. Function tokens such as ':', 'the' and '\n' act as conduits for propagating and re-creating these activations. Prompt 2 shows the same pattern.

The only change between the prompts is replacing 'Russia' with 'UK', while the same function tokens orchestrate different feature combinations, resulting in distinct model outputs.

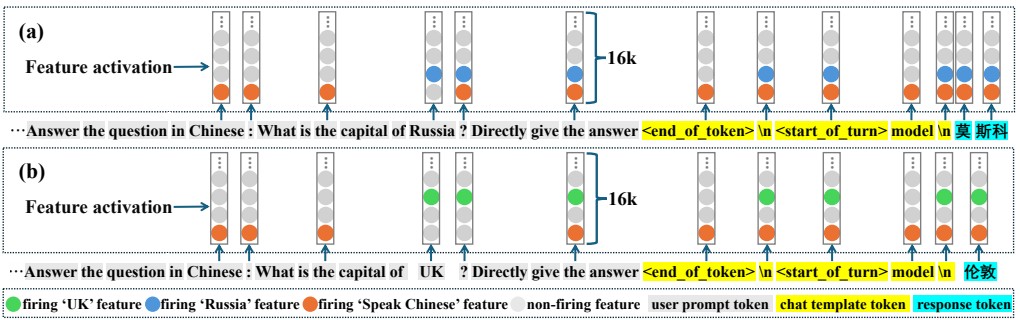

Figure 6: Function tokens can dynamically reactivate predictive features based on different contexts.

| Steer Method | Prompt | Where is Mount Fuji? | Tell me a university. | Could you please recommend a tourist attraction? |
|---|---|---|---|---|
| None | | Japan. | Harvard University | The Eiffel Tower. |
| Activate 'Speak Chinese' feature | | 日本 (Japan) | 哈佛大学 (Harvard University) | 故宫 (The Forbidden City) |
| Activate 'UK' feature | | England | Oxford University | The Eiffel Tower. |
| Activate 'Russia' feature | | Russia | Moscow State University | Alexandrinsky Theatre |
| Activate 'Speak Chinese' feature + Activate "UK" feature | | 英国 | 牛津大学 | 伦敦眼 |
| Activate 'Speak Chinese' feature + Activate "Russia" feature | | 俄罗斯 | 莫斯科国立大学 | 叶卡捷琳娜宫 |

**Chinese Term Translations**

日本: Japan
哈佛大学: Harvard University
故宫: The Forbidden City
英国: UK
牛津大学: Oxford University
伦敦眼: London Eye
俄罗斯: Russia
莫斯科国立大学: Moscow State University
叶卡捷琳娜宫: Catherine Palace

Figure 7: Gemma2-9B-it response when editing activation at the final function token in the prompt.

We show that steering activations on function tokens can directly alter model outputs. The steering method is described in Appendix B. We evaluate this with the following prompts: **Prompt 3**: *Where is Mount Fuji?* **Prompt 4**: *Tell me a university.* **Prompt 5**: *Could you recommend a tourist attraction?* As shown in Figure 7, features activated by the function token are predictive, driving the subsequent token generation. For Prompt 3, the model normally answers in English ('Japan'). Steering only the activations on the final function token in the prompt ('\n') changes the response: activating the 'Speak Chinese' feature switches the answer to '日本' (Japan in Chinese), activating the 'Russia' feature changes the answer to 'Russia', and jointly activating 'Speak Chinese' and 'UK' features yields '英国' (UK in Chinese). Prompts 4 and 5 exhibit the same behavior, demonstrating that function tokens activate predictive features. For more case studies, see Table 11 (§B).

In addition, steering features enable generalized control rather than merely triggering specific word outputs. For example, activating the 'Russia' feature can produce contextually appropriate responses, such as 'Moscow State University' and 'Alexandrinsky Theatre', instead of simply outputting the token 'Russia'. This demonstrates that the features encode high-level semantic concepts.

## 4 MEMORY CONSOLIDATION THROUGH FUNCTION TOKENS

We analyze how memory consolidation occurs during pre-training. We train two models and track their losses on function and content tokens, as well as their feature growth. Our key findings are: (1) the number of the learned features increases with training steps; (2) early training prioritizes predicting function tokens; (3) subsequently, the optimization process becomes dominated by learning to predict content tokens, especially predicting content tokens that follow function tokens.

### 4.1 PRE-TRAINING SETUP

We train two models from scratch using the LLaMA-3.1-8B (Grattafiori et al., 2024) architecture: an 8B model with the originial 32 layers and a 1.5B models with only 2 layers, keeping other components unchanged. We use SlimPajama-627B (Soboleva et al., 2023) as our pre-training corpus, which is a diverse, high-quality collection of web data that has been carefully deduplicated and filtered. This dataset is well-suited for studying memory consolidation during pre-training. We

train for one complete epoch over its 627 billion tokens. We replicate the training hyperparameters of LLaMA-3.1-8B for reproducibility: batch size 1024, max sequence length 4095, AdamW optimizer. The learning rate warm up linearly for 8,000 steps to $8 \times 10^{-5}$, then decays by cosine annealing to $8 \times 10^{-7}$. Training runs on 128 GPUs with 80GB memory each.

## 4.2 MEMORY CONSOLIDATION AS FEATURE EXPANSION

Due to computational constraints, we perform feature decomposition only on the 1.5B model. To track the number of emergent features during pre-training, we train SAEs on second-layer activations at multiple checkpoints. We use JumpReLU-SAE (Rajamanoharan et al., 2024b) with a tanh penalty function (Bloom et al., 2024), wich outperforms alternatives such as TopK-SAE (Gao et al., 2025) and Gated-SAE (Rajamanoharan et al., 2024a). Training details are in Appendix D.

We select three representative checkpoints of the pre-training for SAE training: 3000 steps, 50,000 steps and 130,000 steps, corresponding to early, intermediate and late stages of pre-training. For each checkpoint, we sample text sequences from SlimPajama to obtain 500,000 activations, which are input to the SAE to count the total number of decomposed unique features. As shown in Figure 8, the number of features grows substantially over the progress of pre-training, reflecting the model's increasing representational capability and corresponding to memory consolidation.

Using the bipartite graph analysis described in Section 3.2, we study how token-feature activation evolve. As shown in Figure 8, the number of features grows during training, but function tokens consistently activate most features, in contrast with content tokens. This disparity widens over time, as evidenced by the gradually steepening slopes in the graph.

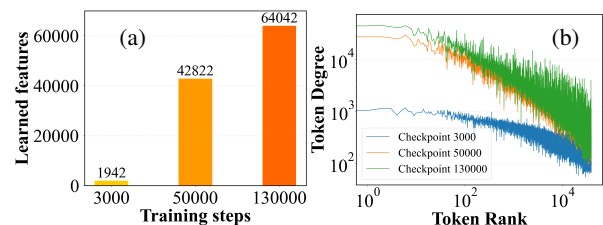

Figure 8: (a) Growth of learned features across training steps. (b) Token degree evolution by checkpoints.

## 4.3 LOSS ON FUNCTION AND CONTENT TOKENS

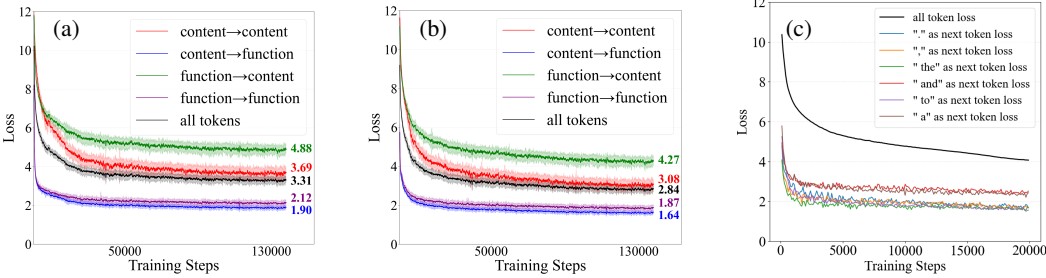

Figure 9: Pre-training loss curves of different token groups.(a) Grouped token loss for 1.5B model. (b) Grouped token loss for 8B model. (c) Typical function token losses in 1.5B model.

To track loss changes for function and content tokens, we categorize next-token prediction of the form $p(\text{next token}|\text{current token}, \text{context})$ into four groups based on whether the current and next tokens are function tokens or content tokens. This yields four distinct categories. For example, $p(\text{next token} = \text{function token} \mid \text{current token} = \text{function token}, \text{context})$ is denoted as function→function. The other three categories are defined similarly: function→content, content→function, and content→content. Figure 9 presents the pre-training loss curves of four groups for both the 1.5B and 8B models, along with the average loss curve across all tokens. We highlight several key observations.

**Function→Content drives the optimization and memory consolidation.** Throughout pre-training, the function→content group has the highest loss in both the 1.5B and 8B models, making it the hardest prediction task. As a result, optimization is dominated by this task, which in turn pushes function tokens to develop the capability to reactivate predictive features from context. Furthermore, the feature growth during pre-training likewise primarily driven by function→content prediction.

**Function token prediction is learned faster and more easily.** For both 1.5B and 8B models, loss decreases more quickly and converge lower when predicting function tokens than content tokens.

Function tokens reach very low loss early in training, showing that LLMs first learn to predict function tokens. Figure 9 plots loss curves of several representative function tokens ('the', 'of', and ',') as next tokens to be predicted, alongside the average loss across all tokens, highlighting rapid convergence within the first 3,000 steps. This indicates that the model first learns to generate function tokens before learning to generate more complex token sequences.

**Scaling enhances content token prediction.** Scaling from 1.5B to 8B parameters yields small loss reductions for content→function group (1.90 to 1.64, $\Delta = 0.26$) and function→function group (2.12 to 1.87, $\Delta = 0.25$), but much larger loss reductions for function→content group (4.88 to 4.27, $\Delta = 0.61$) and content→content group (3.69 to 3.08, $\Delta = 0.61$). These results indicate that scaling model size primarily enhances the content token prediction.

| Model Size | Training Steps | When | young | children | are | learning | to | read | , | they | often | struggle | with | complicated | words |
|---|---|---|---|---|---|---|---|---|---|---|---|---|---|---|---|
| | 100 steps (0.7%) | sharing | SUCCESS | GHz | liament | "). | liament | ブ | Liter | follower | readOnly | readOnly | posites | 투 | になり |
| 1.5b | 3000 steps (2.3%) | , | a | , | a | to | the | the | and | are | , | , | the | and | , |
| | 50000 steps (38%) | the | young | are | in | to | the | the | and | are | find | to | the | ideas | and |
| | 130000 steps (94%) | the | young | are | in | to | read | the | they | are | have | to | the | language | and |
| 8b | 130000 steps (94%) | the | people | are | not | to | read | , | they | are | have | to | reading | words | and |

Figure 10: Next token predictions at different training steps. The first row shows the prompt. Each subsequent row shows the next-token predictions conditioned on all preceding tokens. For example, the third column uses 'When young' as input, and the fourth uses 'When young children' as input.

At last, Figure 10 provides an example to show how LLM generation evolves during pre-training. At the earliest stage (step 100), generation are random. By step 3,000, the model predicts only function tokens (e.g., 'the', 'a'). By step 50,000, it generates locally coherent phrases like 'learning to' and 'to be'. More complex predictions requiring capturing long-range dependencies, emerge at later stages in the 8B model. For instance, correctly predicting the next token after '...struggle with' requires recalling the earlier context 'learning to read'.

## 5 FUNCTION TOKEN HYPOTHESIS

Our experiments suggest that function tokens are crucial for memory consolidation and memory retrieval in LLMs, leading to our Function Token Hypothesis. During inference, function tokens activate the most predictive features from the context to direct the prediction of the next token (memory retrieval). During training, predicting the content token after the function tokens drives parameter updates and feature learning (memory consolidation).

We postulate that the function token hypothesis is the compound result of four factors in LLM training: the training loss (cross entropy loss), learning algorithm (SGD (Ruder, 2017) or backpropagation (Rumelhart et al., 1986)), model architecture (Transformer), and nature of language data.

The training of an LLM is driven by next token prediction. Maximally reducing the loss for next token prediction means making the prediction as accurate as possible. (Minimizing the total loss for predicting all next tokens in the training data is equivalent to compressing the training data as compactly as possible. (Delétang et al., 2024)) During training, the SGD algorithm always manages to reduce the training loss the most by computing and utilizing the steepest descent.

Each block of the Transformer (decoder-only) consists of a multi-head self-attention layer followed by an FFN layer. Both the self-attention layer and the FFN layer can be viewed as key-value memories, as explained. Their roles, however, are different. The self-attention layer is responsible for producing a new internal vector from all internal vectors in the context (note that compositionality is the key characteristic of language (Chomsky, 2002)). The FFN layer is responsible for producing an output vector from the new internal vector. Knowledge is represented as parameters in the FFN layer, and features can be extracted from the output vector.

A natural language text is always segmented by function tokens. From each function token to one of its preceding function tokens, a chunk exists, extending until the beginning of the text. These chunks can represent a phrase, a sentence, or a paragraph, and they are nested. When the LLM's prediction reaches the token immediately following a function token, this implies the start of predicting the next chunk; the task is far more challenging, as it requires understanding the meaning of the entire context

up to that point. This high-challenge prediction compels the LLM to activate the most predictive features in the context during training and reactivate the most predictive features during inference.

Overall, the memory mechanisms of LLMs are extremely complex, due to the complexities of the models and algorithm, as well as the scales of the models and data. Nonetheless, we think that our extensive investigations have convincingly validated the function token hypothesis.

# 6 RELATED WORK

Research on neural memory dates back to the Hopfield network (Hopfield, 1982), also known as associative memory network, which consolidates memories by adjusting weights between neurons. Hopfield networks have evolved into restricted Boltzmann machines (Fischer & Igel, 2012) and feed-forward networks that utilize key-value memories (Geva et al., 2021). Recent research on superposition (Elhage et al., 2022) has shown that it is possible to uncover the features of neural networks such as Transformer, where the number of features is much larger than that of neurons. Through dictionary learning, the superposed activations can be decomposed into monosemantic features. Existing work has demonstrated that such decomposed features can effectively steer model behaviors (Chen et al., 2025b; Panickssery et al., 2024), by maintaining specific feature activations, controlling access to memories, and directing the model through the generation process.

Existing research on LLMs has identified important patterns involving function tokens. For example, separator tokens produce large activations (Sun et al., 2024) and distinct attention weights (Chen et al., 2025a), enabling efficient KV cache designs retaining only separator caches. The crucial role of "formatting" in post-training is also widely recognized (Zhou et al., 2023; Ye et al., 2025; Mamidanna et al., 2025; Li et al., 2025), a function primarily controlled by tokens such as '\n'. Furthermore, recent work on reinforcement learning for reasoning finds that training primarily on high-entropy tokens like 'thus' improves performance (Wang et al., 2025b), while Phi-4 (Abdin et al., 2024) identifies 'pivot tokens', often following function tokens, as critical for response accuracy. We argue these are all function tokens, marked by high frequency and diverse contextual usage. This view is supported by previous work demonstrating that the effective learning of function token representations is crucial for overall LLM performance. Building on this, we propose the Function Token Hypothesis and analyze how these tokens drive memory retrieval and consolidation in LLMs.

# 7 CONCLUSION AND OPEN QUESTIONS

In this work, we propose the function token hypothesis: during inference, function tokens activate the most predictive features from context to guide next token prediction. During pre-training, the prediction of content tokens preceding function tokens drives the model to learn and expand its features. Our experiments provide strong evidence for this hypothesis.

In the meantime, our study raises several open questions:

- One important question is how function tokens acquire the ability to dynamically activate predictive features, in contrast to content tokens. This capability likely emerges from the interplay of model architecture, data nature, training loss, and learning algorithm during training. Investigating this interaction is essential for a better understanding of the phenomena.

- Post-training typically requires only a small number of training steps to achieve substantial improvements in capabilities such as instruction following, chain-of-thought reasoning, and search-agent behavior. Remarkably, training only on function tokens through reinforcement learning can enhance reasoning performance, suggesting that post-training merely activates latent capabilities acquired during pre-training. However, how post-training modifies these activation patterns in function tokens remains an open question.

- In our pre-training experiments, we observe that scaling up (more training data, increased computation, and larger model size) reduces loss, accompanied by an increase in the number of learned features. Notably, function tokens consistently activate most features, exhibiting a scale-free property (token-feature degree distribution follows a power law) throughout training. However, the dynamics of feature formation and the underlying reason of this scale-free property remain unclear, and whether these phenomena follow specific principles requires further investigation.

- Our case studies confirm existing findings that middle layers offer superior interpretability and steerability. However, the mechanistic explanation for why this steerability is concentrated in middle layers, rather than shallow or deep layers, remains elusive.

## ETHICS STATEMENT

We confirm that this work adheres to the ICLR Code of Ethics. Our research aims to understand memory mechanisms in language models, providing insights for building responsible AI systems. All experiments were conducted using publicly available models and datasets, with no use of private or sensitive data.

## REPRODUCIBILITY STATEMENT

To support transparency and reproducibility, we commit to releasing our code upon publication. All experiments were conducted using publicly available models and datasets. We provide comprehensive experimental details in both the main paper and appendix, including model configurations, hyperparameter settings, data preprocessing steps, and evaluation metrics. These measures ensure that our experiments can be fully reproduced by the research community.

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

## A    LLM USAGE DISCLOSURE

We use the large language models (LLMs) to assist with writing in this paper, specifically for spelling and grammar checking, and polishing our text for clarity and readability. All LLM-refined text has been carefully reviewed and verified by the authors to ensure accuracy and alignment with our intended meaning. We did not use LLMs to generate new content, ideas, or experimental results.

## B    STEERING METHOD FOR LARGE LANGUAGE MODELS

Given a target trait, our goal is to identify the corresponding feature in Geema2-9B from the SAE decomposition. Specifically, we extract the feature at the last function token in the prompt, which is an newline token. The process involves four steps:

**Step 1. Collect contrastive prompts.** We construct two sets of prompts: (i) a single prompt that enforces the target trait, and (ii) 20 prompts that do not. For example, to isolate the 'Speak Chinese' feature, we use Prompt 1 ('Answer the question in Chinese: What is the capital of UK?') as the trait-enforcing prompt. This explicitly instructs the model to respond in Chinese. In contrast, Prompt 3 ('Where is Mount Fuji?') is included in the non-trait set, as it contains no language specification and thus defaults to English. The trait-enforcing prompt is used to identify the relevant layer and feature. The non-trait prompts serve as a test set to evaluate steering effectiveness.

**Step 2. Identify the most informative layer.** For each layer $l$, we take the activation of the final function token in the trait-enforcing prompt (e.g., Prompt 1 for 'Speak Chinese') and denote it as a steer vector (Panickssery et al., 2023), $v_l \in \mathbb{R}^d$. We then modify the last function token's activation of each test prompt as $h_l \leftarrow h_l + v_l$, generate responses, and measure the success rate of producing the trait (e.g., 'Speak Chinese'). The layer with the highest success rate is chosen as the most informative. For the traits 'Speak Chinese', 'Russia', and 'UK', the most informative layer all correspond to layer 26.

**Step 3. Identify the feature.** At the chosen layer, we decompose $v_l$ using SAE. By Equation 7, $v_l$ can be expressed by

$$v_l = W_{\text{dec}} \cdot \mathbf{z} + \mathbf{b}_{\text{dec}}, \tag{3}$$

where $\mathbf{z} = (z_1, z_2, \cdots, z_n)^\top$. To locate the trait-specific feature, we rank features by activation strength $z_i$ in descending order. We then apply a binary search to find the smallest $k$ such that activating the top-$k$ features enables the trait, while the top-$(k-1)$ does not. The corresponding steering vector in hidden space is:

$$v_l^{S_k} = \alpha \cdot W_{\text{dec}} \sum_{i \in S_k} \mathbf{e}_i, \tag{4}$$

where $S_k$ is the set of top-$k$ feature IDs, $\mathbf{e}_i$ is the $i$-th standard basis vector, and $\alpha$ is the steering strength. We apply $h_l \leftarrow h_l + v_l^{S_k}$ on the test set and evaluate the success. For 'Speak Chinese', the identified feature is ID 15261 at layer 26; other examples include 'Russia' (feature ID 9591, layer 26) and 'UK' (feature ID 13751, layer 26).

**Step 4. Steering the model.** Once the feature $i$ is identified, the model can be steered with the feature-specific steering vector:

$$v_l^i = \alpha_i \cdot W_{\text{dec}} \mathbf{e}_i. \tag{5}$$

By applying $h_l \leftarrow h_l + v_l^i$ to the last function token of a prompt, we can induce traits such as 'Speak Chinese', 'Russia' or 'UK'.

## C    ADDITIONAL CASE STUDY

We present more interesting examples of steering activations on function tokens, as shown in Figure 11.

| Steer Method | Prompt Please name my newborn baby daughter. | Recommend me a traditional alcoholic beverage. | Recommend me a traditional dish. |
|---|---|---|---|
| None | Eleanor | Sake. | Chicken Tikka Masala |
| Activate 'Speak Chinese' feature | 小雨 (Xiǎo Yǔ) | 啤酒 (Pījiǔ) - Chinese Beer | 意大利面 Carbonara |
| Activate 'UK' feature | Amelia | Scotch whisky. | Shepherd's Pie |
| Activate 'Russia' feature | Anastasia | Vodka Martini. | Borscht. |
| Activate 'Speak Chinese' feature + Activate "UK" feature | 艾莉亚 (Alia) | 威士忌 (Whiskey) | 英国的鱼薯条 |
| Activate 'Speak Chinese' feature + Activate "Russia" feature | 安娜 (Anna) | 伏特加 (Vodka) | 俄式肉丸子 (Russian Meatballs) |

Figure 11: Response of Gemma2-9B-it when editing the activation at the final function token ('\n') in the prompt. The Chinese terms shown in the table and their corresponding English translations are: 小雨 (Xiaoyu, a common Chinese feminine nickname), 啤酒 (beer), 意大利面 (Carbonara), 艾丽娅 (Alia), 威士忌 (Whiskey), 英国的鱼薯条 (British fish and chips), 安娜 (Anna), 伏特加 (Vodka), and 俄式肉丸子 (Russian meatballs).

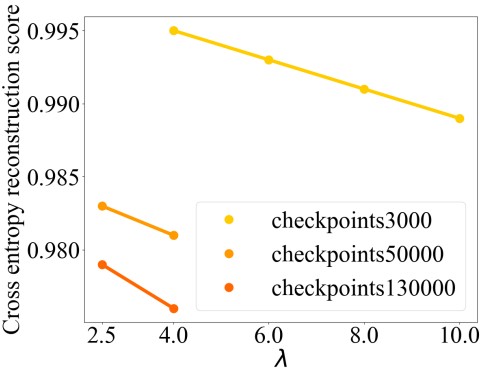

Figure 12: Cross-Entropy reconstruction scores under varying $\lambda$ Values

## D SAE TRAINING DETAILS

Given an activation $\mathbf{x} \in \mathbb{R}^d$ from the residual stream with $n$ dimensions, the JumpReLU-SAE comprises an encoder and decoder:

$$\mathbf{z} = \text{JumpReLU}_\theta(W_{\text{enc}}\mathbf{x} + \mathbf{b}_{\text{enc}}) \tag{6}$$

$$\hat{\mathbf{x}} = W_{\text{dec}}\mathbf{z} + \mathbf{b}_{\text{dec}} \tag{7}$$

where $W_{\text{enc}} \in \mathbb{R}^{n \times d}$, $\mathbf{b}_{\text{enc}} \in \mathbb{R}^n$, $\mathbf{b}_{\text{dec}} \in \mathbb{R}^d$ and $W_{\text{dec}} \in \mathbb{R}^{d \times n}$. The optimization objective combines reconstruction loss with a $L_0$ sparsity penalty:

$$\mathcal{L}(\mathbf{x}) = \underbrace{\|\mathbf{x} - \hat{\mathbf{x}}\|_2^2}_{\mathcal{L}_{\text{reconstruct}}} + \underbrace{\lambda\|\mathbf{z}\|_0}_{\mathcal{L}_{\text{sparsity}}} \tag{8}$$

We train our JumpReLU-SAEs using the open-source library sae_lens[1] (Bloom et al., 2024). Training data consists of one billion activations collected from pre-training dataset (Penedo et al., 2024) with a context size of 1024 tokens. The dictionary width is set to 16 times the activation dimension, resulting in a dictionary size of 65,536. We use a constant learning rate of $1 \times 10^{-5}$.

---

[1]https://github.com/jbloomAus/SAELens

> **Prompt Template**
> <bos><start_of_turn>user
> {prompt} Reply in a single sentence.<end_of_turn>
> <start_of_turn>model

Figure 13: Prompt template

| Steer Method | Where is Mount Fuji? | Tell me a university. | Could you please recommend a tourist attraction? |
|---|---|---|---|
| None | Mount Fuji is located in Japan. | Let me introduce the university of Oxford. | I would recommend you to visit the ancient city of Petra in Jordan. |
| Activate 'Speak Chinese' feature | Mount Fuji is located in日本 (Japan). | Let me introduce the university of清华大学. | I would recommend you to visit the 故宫 (Forbidden City) in Beijing, China. |
| Activate 'UK' feature | Mount Fuji is located in England. | Let me introduce the university of Oxford. | I would recommend you to visit the Tower of London for its rich history and captivating Crown Jewels. |
| Activate 'Russia' feature | Mount Fuji is located in Russia. | Let me introduce the university of Moscow State University. | I would recommend you to visit the Hermitage Museum in Saint Petersburg, Russia. |
| Activate 'Speak Chinese' + 'UK' | Mount Fuji is located in英国. | Let me introduce the university of剑桥 | I would recommend you to visit the 英国伦敦塔, a historic castle with fascinating exhibits and stunning views. |
| Activate 'Speak Chinese' + 'Russia' | Mount Fuji is located in俄罗斯. | Let me introduce the university of莫斯科国立大学. | I would recommend you to visit the 莫斯科红场 (Red Square) in Moscow, Russia. |

Figure 14: Steering activations on different underlined function tokens during model response generation. The Chinese terms shown in the table and their corresponding English translations are:日本 (Japan), 清华大学 (Tsinghua University), 故宫 (Forbidden City),英国 (UK), 剑桥 (Cambridge), 英国伦敦塔 (The Tower of London), 俄罗斯 (Russia), 莫斯科国立大学 (Moscow State University), and 莫斯科红场 (Moscow's Red Square).

We adopt default JumpReLU-SAE training setting: batch size 4096 and a dead-feature (Templeton et al., 2024) detection window of 1000. For JumpReLU, we set the bandwidth to 0.02 and the initialization threshold to 0.01.

SAE training involves a tradeoff between reconstruction quality and sparsity. To quantify reconstruction quality, we use the cross-entropy reconstruction score (Karvonen et al., 2025), which is defined as $\frac{H_* - H_0}{H_{orig} - H_0}$, where $H_{orig}$ is the cross-entropy loss of the original model for next-token prediction, $H_*$ is the cross-entropy loss after substituting the model activation $x$ with its SAE reconstruction during the forward pass, and $H_0$ is the cross-entropy loss when zero-ablating $x$. The metric ranges from 0 to 1, with higher values indicating more faithful reconstruction.

Using the default $L_0$ penalty coefficient, $\lambda = 4$, we find that reconstruction scores varied across early (3,000 steps), intermediate (50,000 steps), and late (130,000 steps) checkpoints, as shown in Figure 12. To make feature counts comparable across these stages, we tuned $\lambda$ to similar reconstruction scores. Specifically, we use $\lambda = 10$ for early checkpoint, $\lambda = 4$ for the intermediate checkpoint, and $\lambda = 2.5$ for the late checkpoint.

From Figure 12, we also observe that, as pre-training progresses, the model's feature representations become increasingly complex and more difficult to decompose.

# E  STEER AT DIFFERENT POSITIONS

In order to allow the model to output more tokens in the response, we modify the prompt template as shown in Figure 13. We select different function tokens in model's responses to steer and observe the following generation.

Figure 14 shows the results of activation steering on function tokens at different positions. Underlines indicate the tokens on which steering is performed. We steer the activations on the tokens 'in', 'of', and 'the' across three different prompts, which are all typical function tokens located at different positions in the response. In the 'Where is Mount Fuji?' example, we steer the activation

of the token 'in'. When the 'speak Chinese' feature is activated, the model can respond in Chinese. When the 'UK' feature is activated, the model tends to provide UK-related responses. Activating multiple features simultaneously on the function token can achieve combinatorial effects. In the response to 'Tell me a university', we manipulate the token 'of', and in the response to 'Could you please recommend a tourist attraction?', we manipulate the token 'the'. Our results demonstrate that steering function tokens is generally effective, not just steer the final newline character.

# F  FUNCTION TOKEN LIST

Table 2 presents all function tokens identified in our experiments, ranked by frequency in SlimPajama-627B in descending order. Tokens not appearing in this table are classified as content tokens.

Table 2: Token statistics with corresponding document coverage, token fractions, and cumulative fractions.

| Token Text | Document Coverage | Token Fraction | Cumulative Fraction |
|---|---|---|---|
| , | 95.00% | 3.60% | 3.60% |
| _the | 90.92% | 3.19% | 6.79% |
| . | 95.80% | 2.31% | 9.10% |
| _and | 89.69% | 1.81% | 10.91% |
| _of | 87.59% | 1.80% | 12.71% |
| _to | 88.71% | 1.68% | 14.40% |
| _ | 81.35% | 1.59% | 15.99% |
| _a | 87.62% | 1.33% | 17.32% |
| _in | 86.04% | 1.16% | 18.48% |
| .\n | 84.58% | 0.91% | 19.39% |
| _is | 78.90% | 0.74% | 20.13% |
| \n | 42.30% | 0.70% | 20.84% |
| _for | 79.82% | 0.64% | 21.48% |
| _that | 67.02% | 0.62% | 22.09% |
| 's | 63.02% | 0.49% | 22.58% |
| _on | 72.40% | 0.47% | 23.05% |
| _with | 73.68% | 0.47% | 23.52% |
| _( | 55.05% | 0.47% | 23.99% |
| : | 52.73% | 0.42% | 24.41% |
| _it | 57.50% | 0.38% | 24.79% |
| _I | 37.43% | 0.38% | 25.17% |
| _as | 61.49% | 0.37% | 25.54% |
| _you | 47.06% | 0.35% | 25.90% |
| _be | 60.03% | 0.33% | 26.23% |
| _are | 60.45% | 0.33% | 26.56% |
| _was | 45.51% | 0.33% | 26.89% |
| 1 | 40.84% | 0.30% | 27.18% |
| _at | 59.38% | 0.29% | 27.48% |
| _by | 58.44% | 0.29% | 27.77% |
| _" | 43.01% | 0.28% | 28.05% |
| _The | 55.12% | 0.28% | 28.34% |
| _from | 61.23% | 0.28% | 28.62% |
| ) | 44.33% | 0.28% | 28.90% |
| _this | 56.27% | 0.26% | 29.16% |
| _have | 55.12% | 0.26% | 29.41% |
| _or | 50.42% | 0.25% | 29.66% |
| 2 | 39.09% | 0.25% | 29.91% |
| - | 38.67% | 0.24% | 30.15% |
| _an | 56.55% | 0.23% | 30.38% |
| 0 | 31.70% | 0.22% | 30.60% |
| | | | Continued on next page |

**Table 2 – continued from previous page**

| Token Text | Document Coverage | Token Fraction | Cumulative Fraction |
|---|---|---|---|
| _not | 46.51% | 0.21% | 30.81% |
| _will | 46.71% | 0.19% | 31.00% |
| _can | 47.99% | 0.19% | 31.19% |
| _has | 49.09% | 0.19% | 31.38% |
| 201 | 33.71% | 0.18% | 31.56% |
| _we | 35.13% | 0.18% | 31.74% |
| \\ | 1.30% | 0.17% | 31.91% |
| The | 48.49% | 0.17% | 32.08% |
| _your | 34.99% | 0.17% | 32.25% |
| 3 | 35.29% | 0.17% | 32.41% |
| _but | 41.84% | 0.16% | 32.57% |
| _his | 25.09% | 0.16% | 32.73% |
| " | 34.19% | 0.16% | 32.88% |
| _all | 45.24% | 0.15% | 33.04% |
| _their | 39.27% | 0.15% | 33.19% |
| _he | 23.69% | 0.15% | 33.34% |
| { | 1.18% | 0.15% | 33.49% |
| _they | 35.37% | 0.15% | 33.64% |
| 't | 33.12% | 0.15% | 33.78% |
| _more | 42.84% | 0.14% | 33.93% |
| _one | 41.94% | 0.14% | 34.07% |
| _which | 40.67% | 0.14% | 34.21% |
| 4 | 31.49% | 0.13% | 34.34% |
| 5 | 32.71% | 0.13% | 34.47% |
| _$ | 12.48% | 0.13% | 34.61% |
| _\ | 0.90% | 0.13% | 34.73% |
| _about | 37.54% | 0.13% | 34.86% |
| --- | 5.40% | 0.11% | 34.97% |
| ; | 21.62% | 0.11% | 35.09% |
| _who | 33.50% | 0.11% | 35.20% |
| _also | 40.22% | 0.11% | 35.31% |
| _our | 30.62% | 0.11% | 35.42% |
| _were | 27.00% | 0.11% | 35.53% |
| _out | 36.49% | 0.11% | 35.64% |
| / | 20.32% | 0.11% | 35.75% |
| 6 | 28.01% | 0.11% | 35.86% |
| _up | 36.43% | 0.11% | 35.97% |
| 8 | 28.60% | 0.11% | 36.08% |
| _been | 35.32% | 0.11% | 36.18% |
| _had | 25.51% | 0.11% | 36.29% |
| _if | 30.49% | 0.10% | 36.39% |
| 7 | 27.31% | 0.10% | 36.50% |
| _so | 33.25% | 0.10% | 36.60% |
| _my | 20.96% | 0.10% | 36.70% |
| _= | 6.62% | 0.10% | 36.80% |
| _time | 34.79% | 0.10% | 36.90% |
| _her | 15.21% | 0.10% | 37.00% |
| 9 | 26.28% | 0.10% | 37.10% |
| _- | 19.91% | 0.10% | 37.20% |
| ' | 27.13% | 0.10% | 37.30% |
| s | 28.83% | 0.09% | 37.39% |
| _would | 27.35% | 0.09% | 37.49% |
| _new | 32.43% | 0.09% | 37.58% |
| _when | 32.82% | 0.09% | 37.67% |
| _other | 33.77% | 0.09% | 37.76% |
| _there | 30.15% | 0.09% | 37.86% |
| | | | Continued on next page |

**Table 2 – continued from previous page**

| Token Text | Document Coverage | Token Fraction | Cumulative Fraction |
|---|---|---|---|
| _A | 28.29% | 0.09% | 37.95% |
| _its | 29.64% | 0.09% | 38.04% |
| _It | 31.56% | 0.09% | 38.13% |
| _like | 30.40% | 0.09% | 38.22% |
| _do | 29.89% | 0.09% | 38.31% |
| _what | 28.23% | 0.09% | 38.39% |
| ---- | 3.87% | 0.09% | 38.48% |
| _, | 18.94% | 0.09% | 38.57% |
| _into | 31.66% | 0.09% | 38.65% |
| 200 | 19.03% | 0.08% | 38.74% |
| } | 2.01% | 0.08% | 38.82% |
| _than | 30.00% | 0.08% | 38.90% |
| _said | 19.12% | 0.08% | 38.98% |
| _some | 29.97% | 0.08% | 39.06% |
| _them | 27.36% | 0.08% | 39.14% |
| _In | 28.39% | 0.08% | 39.22% |
| _& | 17.66% | 0.08% | 39.30% |
| _– | 18.50% | 0.08% | 39.38% |
| _people | 24.05% | 0.08% | 39.46% |
| ing | 29.18% | 0.08% | 39.53% |
| _first | 29.94% | 0.08% | 39.61% |
| )\n | 13.24% | 0.08% | 39.69% |
| I | 23.86% | 0.08% | 39.76% |
| ? | 24.01% | 0.08% | 39.84% |
| A | 27.74% | 0.08% | 39.92% |
| _just | 27.64% | 0.07% | 39.99% |

