# OpenReview forum: "Memory Retrieval and Consolidation in Large Language Models through Function Tokens"
_ICLR.cc/2026/Conference — Submitted to ICLR 2026_

### Official Review · Reviewer_W5Ez · 2025-10-27

**Soundness:** 2
**Presentation:** 3
**Contribution:** 2
**Rating:** 4
**Confidence:** 4

**Summary:**

This paper aims to explain the memory retrieval and consolidation mechanisms of LLMs through a novel "Function Token Hypothesis." The authors first classify tokens based on frequency into "function tokens" (high-frequency, e.g., prepositions, articles, punctuation) and "content tokens" (low-frequency).
The hypothesis contains two core claims:
1. Memory Retrieval (Inference): Function tokens are responsible for activating the most relevant predictive features from the context to govern the generation of subsequent (usually content) tokens.
2. Memory Consolidation (Training): Predicting content tokens that follow function tokens (i.e., function -> content) is a high-difficulty task. The authors posit that the high loss from this task is the primary driver for updating model parameters and learning new features (i.e., consolidating memory).

To support this hypothesis, the authors provide three lines of empirical evidence:

- Bipartite Graph Analysis: Using Sparse Autoencoders (SAEs) to extract features, the authors build a "token-feature" bipartite graph, showing that a few function tokens are connected to (and activate) a vast majority of the model's features (high "degree").
- Activation Steering Case Studies: By causally intervening on the activations of specific features at function token positions (notably \n), they can change the model's output (e.g., factual content or language).
- Pre-training Dynamics Analysis: By training models from scratch and decomposing the loss, they show that the function -> content prediction loss is consistently the highest throughout training.

**Strengths:**

1. Novel Analytical Perspective: The paper's primary strength is its originality in proposing a new analytical framework. By classifying tokens into "function tokens" and "content tokens" based on frequency, the authors provide a novel and intuitive lens to investigate the long-standing question of LLM memory mechanisms. This perspective itself is a valuable contribution, opening a new direction for interpretability research.

2. Comprehensive Analysis with Novel Methods: The paper systematically analyzes the role of these function tokens across both key phases of a model's lifecycle:
	- During Inference (Retrieval): The authors introduce a novel method using SAE-extracted features to build a token-feature bipartite graph. This approach provides a quantitative measure (token degree) to support their claim that function tokens act as "hubs" for feature activation.
	- During Pre-training (Consolidation): The paper also investigates the training process by tracking the loss dynamics of different token-type transitions (e.g., function -> content). This is a compelling method for analyzing how such mechanisms might be learned.

**Weaknesses:**

The paper’s inferential leap from its insightful observations to its central hypothesis lacks rigor at several key points, primarily by conflating correlation with causation.

1. Linguistic Priors as an Alternative Explanation for High Token Degree: The paper argues that the high 'token degree' of function tokens (Fig 5) proves they are "responsible" for retrieval. However, this observation can be more simply explained as a direct artifact of a fundamental linguistic prior:

	- From a linguistic and informational standpoint, function tokens (e.g., "in") naturally precede a much wider and more uncertain range of subsequent tokens than content tokens (e.g., "Rowling").

	- This high contextual diversity mechanically requires function tokens to be associated with a more diverse set of features during training.

	- Therefore, the high 'degree' is a statistical correlation that confirms this linguistic property, but it does not, by itself, prove that function tokens are causally responsible for activating memory in any specific instance.

2. Correlation vs. Causation in the Training Loss Argument: This same linguistic prior also undermines the "memory consolidation" claim.

	- The paper observes that the function -> content loss is highest (Fig 9) and concludes this drives optimization, which in turn "pushes function tokens to develop the capability to reactivate..."

	- This is a logical leap. The high loss is an expected consequence of the task's inherent difficulty (high uncertainty/entropy). While high loss certainly contributes to the optimization gradient, the paper fails to prove a causal link—that this pressure specifically results in the function token becoming a "hub." It confuses the cause (inherent task difficulty) with the proposed, unproven effect (a specific hub-like capability developing in the function token).

3. Confounding Variables in the Activation Steering Experiment: The causal evidence provided (Section 3.2, Fig 6-7) is undermined by a critical flaw in its experimental design.

	- The experiment confounds two variables: "token type" (function) and "token position" (closest to the answer).

	- In a Transformer, the next-token prediction is naturally most influenced by the final hidden state of the last token. The authors intervene on the last token (\n), which happens to also be a "function token."

	- This makes it impossible to know if the effect is due to the token's functional property (as hypothesized) or its positional property (the standard mechanism). A rigorous control experiment, such as showing that a distant function token has more influence than a closer content token, is missing.

4. Lack of Practical Utility and Exploitability: The hypothesis is presented as a pure observation, but the paper fails to demonstrate its practical utility.

	- A stronger contribution would leverage this insight to propose tangible improvements. For instance, the paper does not explore:

		- (a) If this hypothesis can be used to improve training (e.g., by re-weighting the function -> content loss)?

		- (b) If it provides a new method for error detection (e.g., finding faulty activations at function tokens)?

		- (c) If it enables more robust inference-time control?

	- Without this, the hypothesis remains an "interesting phenomenon" rather than a "useful theory."

**Questions:**

1. The paper defines function tokens using frequency. This definition leads to the inclusion of different types of tokens, such as (a) punctuation/delimiters, (b) grammatical words (prepositions/articles), and (c) others (e.g., numbers). Is the behavior of these different token types expected to be uniform under the paper's hypothesis? Or might there be differences? Is it possible that punctuation/delimiters, rather than all tokens in the list, might be dominating the observations?

2. Regarding the confounding variables in the steering experiment (Weakness 3), it's unclear whether the observed effect is due to token type (function) or position (last token). It would be insightful to see these variables disentangled. For example, exploring the effect of steering on a distant function token versus a closer content token could be a valuable direction. The hypothesis in this paper would predict the former to be more effective, and further investigation in this direction could significantly strengthen the claim.

---

> ### Author Response · Authors · 2025-11-20
> **Rebuttal by Authors Part1**
>
> We sincerely thank you for the time and effort you invested in reviewing our paper. We greatly appreciate your thoughtful comments and your recognition of our work in providing a novel perspective for interpretability research and a comprehensive empirical analysis. We have carefully addressed your concerns as detailed below.
>
> ---
> > **Q1: Linguistic Priors as an Alternative Explanation for High Token Degree: The paper argues that the high 'token degree' of function tokens (Fig 5) proves they are "responsible" for retrieval. However, this observation can be more simply explained as a direct artifact of a fundamental linguistic prior.**
>
> A1: Thank you for raising this point. We would like to clarify our position and address your concern:
> - Our paper does not claim that the high token degree of function tokens proves they are responsible for retrieval.
> - To the best of our knowledge, Figure 5 provides the first practical evidence (via the large-scale token-feature bipartite graph) that function tokens are able to activate most features in LLMs. This is an empirical observation rather than a theoretical assumption.
> - This phenomenon cannot be fully attributed to linguistic priors alone. In the paper, we explicitly discuss that multiple factors may contribute to this behavior, not only linguistic prior, but also training loss, learning algorithms, and model architecture. For example, it is unclear whether a BERT model, which shares the same linguistic priors, would exhibit the same pattern.
>
> ---
> > **Q2: Correlation vs. Causation in the Training Loss Argument: This same linguistic prior also undermines the "memory consolidation" claim.**
>
> A2: Thanks for your thoughtful feedback. We would like to clarify our position and address your concern regarding the interpretation of Figure 9 and the memory consolidation hypothesis.
> - Our memory consolidation hypothesis states that during pre-training, predicting content tokens based on the function tokens drives the LLM to update its parameters to learn and expand features. This hypothesis is supported by the pre-training loss curves shown in Figure 9. We agree with you that the high loss of function->content task reflects its inherent difficulty, and therefore contributes most to the optimization gradient. These interpretations are consistent and jointly explain the empirical phenomenon observed in Figure 9.
> - To the best of our knowledge, our work is the first to systematically track and analyze pre-training loss curves of LLMs from the perspective of function/content token. We provide solid statistical evidence showing that the four types of next-token prediction tasks (function->function, function->content, content->function, content->content) exhibit distinct loss dynamics. We believe this represents an insightful and novel observation, offering new directions for improving pre-training efficiency and interpretability in future work.
> - Regrading the claim that the function->content task pushes is the function tokens to develop the capability to reactivate predictive features from context, we present it as a reasonable, hypothesis-generating interpretation, not as a definitive causal claim. Our goal is to stimulate further research on how function tokens contribute to the feature formation during pre-training.
> - In summary, the main contribution of the memory consolidation hypothesis are: (1) We design and conduct carefully experiments in large language models demonstrating that the number of learned features increases with training steps; (2) We introduce a novel loss-tracking method to demonstrate that the dynamics of pre-training: early training prioritizes predicting function tokens; subsequently, the optimization process becomes dominated by learning to predict content tokens, especially predicting content tokens that follow function tokens.
>
> We hope these clarifications convey that our primary contribution lies in methodological innovation and solid empirical observation, rather than asserting a strict causal claim. Our aim is to provide interpretable tools and insights to advance understanding of pre-training dynamics in large language models.

---

> ### Author Response · Authors · 2025-11-20
> **Rebuttal by Authors Part2**
>
> > **Q3: Confounding Variables in the Activation Steering Experiment: The causal evidence provided (Section 3.2, Fig 6-7) is undermined by a critical flaw in its experimental design.**
>
> A3: To determine whether the observed effect due to the token being a function token or due to its position, we conduct additional experiments in which we steer activations on other function tokens that are not the final position. The results are shown in Figure 13 in Appendix E in the revised paper. The results show that steering activations on tokens such as 'in', 'of' and 'the' consistently alters the model's output.
> For example, when we steer the token 'in' within the response by manually activating the 'Speak Chinese' and 'UK' features, the model's answer to 'Where is Mount Fuji?' shifts from 'Mount Fuji is located in Japan.' to 'Mount Fuji is located in英国'. This demonstrates that the effect is not restricted to the last-position token and that steering other function tokens can influence the generated text.
>
> ---
> > **Q4: Lack of Practical Utility and Exploitability: The hypothesis is presented as a pure observation, but the paper fails to demonstrate its practical utility.**
>
> A4: Thanks for this insightful comment. As discussed in our paper, recent works have shown that training strategies focusing on certain "special tokens" can enhance model behavior. For example, the use of "pivot tokens" during post-training can significantly enhance performance in response [1], and training that concentrates on high-entropy tokens also yields better performance [2]. We argue that these tokens are all function tokens. Our work provides a unifying interpretability framework that explains why these methods are effective from the perspective of the Function Token Hypothesis.
> Beyond connecting to existing techniques, we agree with you that the function token hypothesis opens promising directions for practical applications, including training improvement, error detection, and robust inference-time control. We will to explore these in future work.
> Finally, we would like to emphasize that establishing a significant and reproducible phenomenon itself constitutes a valuable scientific contribution. Our work conducts the first large-scale analyses of token-feature degree distributions, feature expansion during pre-training, and loss trajectories across training stages. The Function Token Hypothesis offers a coherent interpretability framework that connects these empirical observations into a unified understanding.
>
> [1] Phi-4 Technical Report.
>
> [2] Beyond the 80/20 Rule: High-Entropy Minority Tokens  Drive Effective Reinforcement Learning for LLM Reasoning.
>
> > **Q5: The paper defines function tokens using frequency. This definition leads to the inclusion of different types of tokens, such as (a) punctuation/delimiters, (b) grammatical words (prepositions/articles), and (c) others (e.g., numbers). Is the behavior of these different token types expected to be uniform under the paper's hypothesis? Or might there be differences? Is it possible that punctuation/delimiters, rather than all tokens in the list, might be dominating the observations?**
>
> A5: Thanks for your question. Our definition of function tokens is based on frequency, so it naturally includes punctuation, grammatical words, and some others. Despite their surface differences, the most frequent tokens show similar functional behavior in the model because they share comparable frequencies and broad contextual distributions. (Note that numbers are much less frequent and appear in narrower contexts) For example, as shown in Figure 5, all high-frequency tokens can access most features on the token-feature bipartite graph. Figure 6 shows that not only punctuation/delimiters but also other function token (e.g., 'the') can reactivate predictive features from context, suggesting that the observed effect is not dominated by punctuation alone.
>
> ---
> > **Q6: Regarding the confounding variables in the steering experiment (Weakness 3), it's unclear whether the observed effect is due to token type (function) or position (last token). It would be insightful to see these variables disentangled. For example, exploring the effect of steering on a distant function token versus a closer content token could be a valuable direction. The hypothesis in this paper would predict the former to be more effective, and further investigation in this direction could significantly strengthen the claim.**
>
> A6: In response to Q3, we perform an experiments demonstrating that the capability to reactivate the most predictive features from context is due to the token type (function token), rather than its position.

---

> ### Author Response · Authors · 2025-11-26
>
> Thank you for taking the time to review our manuscript. We sincerely appreciate your insightful feedback. As the deadline for the discussion phase approaches, we would like to kindly remind you of our rebuttal, in which we diligently addressed each concern you raised in your feedback.
> If you have any further questions or concerns, we would be happy and eager to address them promptly in the discussion period.
> Thanks again for your valuable feedback.

---

> ### Comment · Reviewer_W5Ez · 2025-11-28
>
> I thank the authors for their detailed rebuttal and the additional experiments included in Appendix E. However, after carefully examining the new results and the clarifications regarding the hypothesis, I find that my core concerns have not been resolved.
>
> ### Regarding Q1 (Linguistic Priors vs. Mechanism):
> While I appreciate the authors' acknowledgment that the high degree of function tokens is an empirical observation, the rebuttal fails to convincingly dissociate this phenomenon from simple linguistic priors.
> * Lack of Comparative Evidence: The claim that this pattern is driven by "training loss and model architecture" rather than just data statistics remains unsubstantiated. The authors admit it is "unclear" if non-autoregressive models (like BERT) would exhibit this pattern. Without comparing causal LLMs against architectures that do not rely on next-token prediction, there is no evidence to rule out that *any* model trained on natural language would statistically mirror these linguistic priors.
>
> ### Regarding Q2 (Predictability of Loss Patterns):
> I find the authors' response unconvincing regarding the novelty and supportive power of the reported phenomenon. The observation that the Function -> Content prediction task yields the highest loss (Figure 9) is entirely predictable and expected, rather than a novel discovery that supports a specific "memory consolidation" mechanism.
> * Information Theoretic Inevitability: Linguistically, function tokens (e.g., 'the', 'of') precede a vast and open-ended set of content tokens. This transition inherently possesses the highest entropy. Mathematically, high entropy directly translates to high cross-entropy loss. Observing this is simply confirming a fundamental statistical property of natural language, not identifying a unique learning mechanism of the model.
>
> ### Regarding Q3 (Confounding Variables in Steering Experiments):
> I appreciate the effort to conduct additional experiments in Appendix E (Figure 14). However, these new results fail to resolve the core concern regarding the confounding variable of "position" (specifically, immediate adjacency).
>
> 1. Persisting Confounding Variable:
> In all three new examples provided in Figure 14, the steered function tokens (in, of, the) are the immediate predecessors of the target content tokens (Japan, Tsinghua, Forbidden City). In autoregressive Transformers, the token immediately preceding the prediction target naturally dominates the information flow. Therefore, showing that steering the immediate predecessor controls the next token confirms the standard "next-token prediction" mechanism rather than a specialized "retrieval function" unique to function tokens.
>
> 2. Missing Crucial Controls:
> To truly validate the "Function Token Hypothesis" against a positional baseline, the experimental design requires rigorous controls which are currently absent:
>
> * Control A (Token Type - The "Content Token" Test): You must demonstrate that steering a content token that is an immediate predecessor fails to control the output.
>     * Specific Example: In the sequence ...university of Tsinghua [University], steer the content token "Tsinghua" with the "Speak Chinese" feature.
>     * Hypothesis Test: If steering the content token "Tsinghua" successfully forces the model to generate the Chinese equivalent "大学" instead of "University," it proves that content tokens are equally capable of governing next-token prediction when they are in the immediate preceding position. This would suggest the phenomenon is driven by proximity, not by the unique nature of function tokens.
>
> * Control B (Distance): You must demonstrate that steering a distant function token (e.g., a function token early in the prompt, far from the target generation) can effectively govern the retrieval of the specific fact. If the hypothesis holds that function tokens act as global retrieval hubs, this distant steering should be effective.
>
> Conclusion:
> The current experimental evidence only proves that "the last token influences the next token," which is a known property of the architecture. The unique role of function tokens remains unproven due to the lack of distance and type controls.
>
>
> Consequently, I am inclined to lower my rating.

---

> > ### Author Response · Authors · 2025-12-03
> > **Final Response to Reviewer W5Ez - part1**
> >
> > > Regarding Q1 (Linguistic Priors vs. Mechanism):
> > > While I appreciate the authors' acknowledgment that the high degree of function tokens is an empirical observation, the rebuttal fails to convincingly dissociate this phenomenon from simple linguistic priors.
> > > - Lack of Comparative Evidence: The claim that this pattern is driven by "training loss and model architecture" rather than just data statistics remains unsubstantiated. The authors admit it is "unclear" if non-autoregressive models (like BERT) would exhibit this pattern. Without comparing causal LLMs against architectures that do not rely on next-token prediction, there is no evidence to rule out that any model trained on natural language would statistically mirror these linguistic priors.
> >
> > ---
> >
> > Thanks for your feedback. Our earlier response may not have clearly conveyed our intention, which was simply to note that the statement in your comment "this observation can be more simply explained as a direct artifact of a fundamental linguistic prior" is not sufficiently precise.
> > The main contribution of the work is the Function Token Hypothesis, which proposes that function tokens can reactivate most predictive features from the context. We support this claim by large-scale token-feature bipartite graph analysis and case studies on steering behavior on function tokens. These results highlight a mechanistic property of how LLM internally use function tokens at the feature level.
> > We agree that understanding why this phenomenon emerges is an intriguing and open question. However, the value of our contribution does not depend on fully resolving this causal origin. The formulation of the phenomenon, together with our new analytical perspective for studying model interpretability, is substantive and should no be overlooked.
> >
> > ---
> >
> > > Regarding Q2 (Predictability of Loss Patterns):
> > > I find the authors' response unconvincing regarding the novelty and supportive power of the reported phenomenon. The observation that the Function -> Content prediction task yields the highest loss (Figure 9) is entirely predictable and expected, rather than a novel discovery that supports a specific "memory consolidation" mechanism.
> > > - Information Theoretic Inevitability: Linguistically, function tokens (e.g., 'the', 'of') precede a vast and open-ended set of content tokens. This transition inherently possesses the highest entropy. Mathematically, high entropy directly translates to high cross-entropy loss. Observing this is simply confirming a fundamental statistical property of natural language, not identifying a unique learning mechanism of the model.
> >
> > ---
> >
> > Thanks for your comment. While we agree that in a context-free setting, transition from function tokens to content tokens naturally possess high entropy, we should note that with context (the case for LLM), the next token following a function token can be predictable. For example, "The capital of France is __", the next token following "is" has extremely low entropy in the corpus.
> > Thus, the high loss of function->content cannot be attributed to linguistic entropy. Instead, it reflects the model's difficulty in recalling or integrating the relevant information at that point in training. This precisely reflects memory consolidation.

---

> > ### Author Response · Authors · 2025-12-03
> > **Final Response to Reviewer W5Ez - part2**
> >
> > > Regarding Q3 (Confounding Variables in Steering Experiments):
> > I appreciate the effort to conduct additional experiments in Appendix E (Figure 14). However, these new results fail to resolve the core concern regarding the confounding variable of "position" (specifically, immediate adjacency).
> > > 1. Persisting Confounding Variable: In all three new examples provided in Figure 14, the steered function tokens (in, of, the) are the immediate predecessors of the target content tokens (Japan, Tsinghua, Forbidden City). In autoregressive Transformers, the token immediately preceding the prediction target naturally dominates the information flow. Therefore, showing that steering the immediate predecessor controls the next token confirms the standard "next-token prediction" mechanism rather than a specialized "retrieval function" unique to function tokens.
> > > 2. Missing Crucial Controls: To truly validate the "Function Token Hypothesis" against a positional baseline, the experimental design requires rigorous controls which are currently absent:
> > > - Control A (Token Type - The "Content Token" Test): You must demonstrate that steering a content token that is an immediate predecessor fails to control the output.
> > >  - Specific Example: In the sequence ...university of Tsinghua [University], steer the content token "Tsinghua" with the "Speak Chinese" feature.
> > >  - Hypothesis Test: If steering the content token "Tsinghua" successfully forces the model to generate the Chinese equivalent "大学" instead of "University," it proves that content tokens are equally capable of governing next-token prediction when they are in the immediate preceding position. This would suggest the phenomenon is driven by proximity, not by the unique nature of function tokens.
> > > - Control B (Distance): You must demonstrate that steering a distant function token (e.g., a function token early in the prompt, far from the target generation) can effectively govern the retrieval of the specific fact. If the hypothesis holds that function tokens act as global retrieval hubs, this distant steering should be effective.
> >
> > ---
> > First, we would like to clarify the motivation and setting of our experiments.
> > - The experiments aim to demonstrate that function tokens can reactivate the most predictive features from the context to predict the next token. Therefore, our primary focus is on how the next-token changes when a function token is steered. Whether the representation of function tokens will continue to influence the generation of distant next tokens is a separate question.
> > - Some reviewers raised a concern that in the earlier version of our experiments, the function token is also the final token of the prompt. To address this, we provided supplementary experiments that the function token is at arbitrary positions within the model's response rather than only at the end of the prompt.
> > - The results shown that steering function tokens (whether placed in the prompt or response), can effectively alter the next token prediction, supporting the function token hypothesis.
> >
> > Second, we provide additional experimental results on steering a content token as below. These results indicate that steering on a content token has limited impact on the model's subsequent generation.
> > - Regarding the reviewer's experiments "Specific Example: In the sequence ...university of Tsinghua [University], steer the content token "Tsinghua" with the "Speak Chinese" feature."
> >   - If we feed “...university of Tsinghua” to LLM，model is unlikely to generate [University] as the next token，since "university of Tsinghua University"is not a fluent expression.
> >   - We used the same steering method and steering strength as in Appendix E, only change the steer token from "of" to "university" with "Speak Chinese feature"，and the model generated “Let me introduce the university：Harvard University”
> > - We conducted large-scale statistics on the relationship between different tokens and feature activations in the bipartite graph experiments (Figure 5). It can be clearly observed that function tokens have the potential to activate various different features, while content tokens can only activate a limited set of features. Therefore, we propose the function token hypothesis: function tokens activate the most predictive features according to the context, where some predictive feature may be selectively aggregated through and others may be through MLPs that generate new activations. Therefore, it is reasonable to steer the certain feature on function tokens, because most features can potentially be activated on function tokens, but may not necessarily be activated on content tokens.

---

### Official Review · Reviewer_XS8e · 2025-10-30

**Soundness:** 3
**Presentation:** 2
**Contribution:** 1
**Rating:** 2
**Confidence:** 4

**Summary:**

The authors find that some common tokens (that they call "function tokens") activate SAE features. They conduct experiments to examine how SAE feature activations change at function tokens based on context, and they examine the behavior of function tokens vs context tokens during pre-training.

**Strengths:**

This paper attempts to formalize some folk wisdom that models will aggregate information on common tokens (e.g. prepositions, punctuation).

A particular strength is the examination of model behavior during pre-training. My (incomplete) understanding is that models learn n-gram statistics fairly early during training, then start to generalize beyond that. It would be interesting to see if models start aggregating information on "function" tokens as a way to transition past the n-gram regime.

**Weaknesses:**

- I think it's fairly well known that language models will use common tokens (e.g. punctuation, prepositions) to aggregate information from previous parts of the context, including some previous mechanistic work on understanding how models do this (e.g. Dissecting Recall of Factual Associations in Auto-Regressive Language Models).
- Throughout the paper, there seems to be a confusion about what is going on; the "function tokens" are almost certainly not activating the features, but rather aggregating features from previous entities. So, while the SAE feature activation is happening in the residual stream associated with the "function token," it seems kind of misleading to say that the function tokens are activating these features.
- Classifying function tokens simply based on frequency in the training data doesn't seem like a terribly principled way to decide what's a function token.
- The comments on the loss of the function tokens during training seem kind of obvious? The model is better at predicting the most common tokens?

A few directions that could be interesting:
- Is there a principled way to figure out which tokens tend to aggregate information?
- I think the training dynamics and transitions in model behavior during pretraining (see comment above) related to function tokens is particularly interesting.

Nitpick:
- line 93: I've seen papers where people are loose with the term "neuron," but this generally refers to a weighted linear combination plus activation function (e.g. input into the FFN) — not just an axis in the residual stream

**Questions:**

Overall, I found the bipartite graph section to be fairly confusing. I think I get it, but just to be sure: is the idea that we are collecting all the SAE features that activate on the residual stream for a specific token in the SlimPajama dataset?

---

> ### Author Response · Authors · 2025-11-20
> **Rebuttal by Authors Part1**
>
> We sincerely thank you for your effort in reviewing our paper. We greatly appreciate your valuable comments and feedbacks. In our response, we will address each question individually, quoting them and providing our answer accordingly. Please let us know if our responses address your concerns.
>
> ---
> > **Q1: I think it's fairly well known that language models will use common tokens (e.g. punctuation, prepositions) to aggregate information from previous parts of the context, including some previous mechanistic work on understanding how models do this (e.g. Dissecting Recall of Factual Associations in Auto-Regressive Language Models).**
>
> A1: The work [1] primarily focuses on studying factual associations by analyzing subject-relation queries. In contrast, our work differs both in motivation and methodology. We aim to explain general memory retrieval in LLMs from the perspective of function tokens, focusing on the overall memory mechanisms rather than factual knowledge alone. Moreover, our analysis is based on the model's feature representation space, providing a more fundamental, representation-level understanding of memory mechanisms.
> While it may seem intuitive that LLMs use tokens like punctuation or prepositions to aggregate contextual information, this observation, though correct, is not sufficient. There remains a gap: why and under what conditions such tokens exhibit this behavior. Moreover, the underlying mechanistic interpretation of how this retrieval occurs has not be explored. Our work systematically investigates these questions. In particular:
> - The Function Token Hypothesis proposes that function tokens can reactivate the most predictive features from context to guide the next token prediction. Note that our identification of function token is not  based on linguistic categories (e.g., punctuation, prepositions), but rather on token frequency in the training corpus. Since the model has no inherent linguistic knowledge, tokens with similar distribution are functionally equivalent from the model's perspective. Thus, frequency provides a more intrinsic criterion for identifying function tokens.
> - We perform systematic empirical experiments to support the hypothesis. Token-feature bipartite graph analysis shows that a small number of function tokens can access to most features. This finding shows intrinsic distinction between function and content tokens.
> - Pre-training dynamics analysis demonstrates that the optimization process is dominated by function->content prediction tasks, which drives function tokens to develop the ability to reactivate contextually predictive features.
> - Our case study further trace feature activations on a token-by-token manner and demonstrate that the same function token can activate different features depending on the context. To the best of our knowledge, we are the first to report these empirical findings.
>
> [1] Dissecting Recall of Feature Associations in Auto-Regressive Language Models. EMNLP 2023.
>
> ---
>
> > **Q2: Throughout the paper, there seems to be a confusion about what is going on; the "function tokens" are almost certainly not activating the features, but rather aggregating features from previous entities. So, while the SAE feature activation is happening in the residual stream associated with the "function token," it seems kind of misleading to say that the function tokens are activating these features.**
>
> A2: We would like to clarify the mechanism. Feature activation refers to that when layer activations are input into the SAE, certain features become active. This indicates that those features are present in the residual stream at that position. As illustrated in Figure 6, the function token can reactivate predictive features that appeared earlier in the context, reflecting a memory retrieval process. Note that , besides reactivating contextual features, function tokens can also activate new features that were not present in the preceding context, through the FFN . As shown in Figure1, the token "in" can also activate the feature "england", which has not appeared in the previous text. This is most likely because the combination of lower-level features triggered a new feature activation. We plan to investigate further in future work.
>
> ---
> > **Q3: Classifying function tokens simply based on frequency in the training data doesn't seem like a terribly principled way to decide what's a function token.**
>
> A3: As discussed in A1, our identification of function tokens is not based on predefined linguistic categories. Instead, it relies on token frequency in the training corpus. Since the model has no access to explicit category labels, it learns purely from statistical patterns in token usage. From the model's perspective, tokens that share similar frequency and contextual distributions in the training data play comparable roles. Therefore, frequency provides a more intrinsic and effective criterion for identifying function tokens.

---

> ### Author Response · Authors · 2025-11-20
> **Rebuttal by Authors Part2**
>
> > **Q4: The comments on the loss of the function tokens during training seem kind of obvious? The model is better at predicting the most common tokens?**
>
> A4: Thanks for your comments. We agree that the model shows stronger performance in predicting function tokens. However, Figure 9 aims to emphasize a non-trivial and insightful finding: the function->content drives the optimization process. This observation not only explains why function tokens can retrieve predictive features from the context, but also why function tokens have access to most features on the token-feature bipartite graph. In addition, we also demonstrate that scaling mainly enhances content token prediction.
>
> ---
> > **Q5: line 93: I've seen papers where people are loose with the term "neuron," but this generally refers to a weighted linear combination plus activation function (e.g. input into the FFN) — not just an axis in the residual stream**
>
> A5: Thank you for raising this point. We have revised our paper to ensure precise.
>
> ---
>
> > **Q6: Overall, I found the bipartite graph section to be fairly confusing. I think I get it, but just to be sure: is the idea that we are collecting all the SAE features that activate on the residual stream for a specific token in the SlimPajama dataset?**
>
> A6: During step (2) decompose feature step (Line 221), for each token activation, we collect all SAE features it activates. Note that, the same token may activates different SAE features depending on its context. Then, in step (3) build bipartite graph, a token is linked to a feature if it activates the feature in any context.

---

> ### Author Response · Authors · 2025-11-26
>
> Thank you for taking the time to review our manuscript. We sincerely appreciate your insightful feedback. As the deadline for the discussion phase approaches, we would like to kindly remind you of our rebuttal, in which we diligently addressed each concern you raised in your feedback.
> If you have any further questions or concerns, we would be happy and eager to address them promptly in the discussion period.
> Thanks again for your valuable feedback.

---

> > ### Comment · Reviewer_XS8e · 2025-11-27
> >
> > I agree that there is probably something interesting happening with common tokens aggregating (and extracting) features, but this is already well-known. Setting a (somewhat arbitrary) threshold for token frequency doesn't seem to offer much additional insight. Some directions that I think are promising: when and how does this behavior evolve during training? Can you understand more explicitly how feature aggregation happens?

---

> > > ### Author Response · Authors · 2025-11-27
> > >
> > > Thanks for your feedback. We would like to clarify that our contribution goes beyond defining what is function token. Our main contributions include:
> > > - Through feature decomposition in LLM, we discover that only a very small number of function tokens can activate nearly all features in the model, which represents a novel scientific observation.
> > > - We perform experiments showing how function tokens reactivate predictive features from context. To our knowledge, no prior work has observed this phenomenon at the feature level.
> > > - By tracking the pre-training, we demonstrate that the model learning is largely driven by function->content prediction.
> > >
> > > Regarding your question of "when and how this behavior evolves during training", we have conducted feature decompositions at multiple stages of LLM pre-training (**Figure 8(b)**). The results show that function tokens consistently access most features, indicating that this behavior is a persistent pattern rather than a stage-specific effect.
> > >
> > > We agree that understanding how feature aggregation happens is important, but technically very challenging. Our work offers a new and effective perspective of function tokens for this challenging question, and we leave this as future work.
> > >
> > > At last, although prior works have noted that some special tokens play important roles during training, to the best of our knowledge, we are the first to formally identify them as function tokens and to demonstrate that this property is primarily determined by token distribution in the training corpus, rather than by any linguistic definition.
> > >
> > > The 40% partition is not arbitrary. It is chosen to balance coverage and purity. If we extend the threshold, for example to 50%, admits 300 additional token types, many of which have extremely low document coverage (<10%). These tokens appear frequently only in a small subset of documents in the training corpus, reflecting the "bursty" phenomenon shown in Figure 3(b). Including them would dilute the set and undermine its interpretability as a group of broadly distributed, high-frequency tokens. To preserve a cleaner function token set, we use the 40% threshold.
> > >
> > > Thanks again for engaging in this discussion. We appreciate your time and effort on reviewing our work.

---

### Official Review · Reviewer_JC5g · 2025-10-30

**Soundness:** 2
**Presentation:** 3
**Contribution:** 3
**Rating:** 6
**Confidence:** 3

**Summary:**

This paper proposes the Function Token Hypothesis to explain memory mechanisms in LLMs. The authors define function tokens as the 122 most frequent tokens (covering 40% of corpus occurrences), primarily punctuation, articles, and prepositions, and contrast them with content tokens. The hypothesis claims that during inference, function tokens activate predictive features from context to guide next-token prediction (memory retrieval), while during pre-training, predicting content tokens after function tokens drives parameter updates and feature expansion (memory consolidation).The evidence comes from three experiments: (1) bipartite graph analysis on Gemma2-9B showing the top 10 function tokens activate 76% of SAE-decomposed features, (2) steering experiments showing that modifying feature activations at function token positions controls model outputs, and (3) pre-training experiments showing function -> content predictions have the highest loss and dominate optimization.

**Strengths:**

The bipartite graph analysis is creative and novel - mapping 128K tokens to 965K SAE features at scale provides a unique lens for understanding token-feature relationships. Additionally, the steering experiments are convincing as they combine multiple features (“Speak Chinese” + “UK” producing Chinese text about the UK). Overall, the experimental approach from the large-scale statistical analysis to pre-training analysis seemed sound and convincing.

**Weaknesses:**

The core definition conflates frequency with linguistic function, making it impossible to determine whether or not effects arise from frequency effects. No experiments match tokens by frequency or justify the arbitrary 40% threshold. Both the choice of 40% threshold and 122 tokens seems arbitrary, and probably contains tokens that would be classified as content tokens in linguistics but are included in their list.

Only three features are analyzed in detail in section 3.2 with no statistics across all features.

The studies are limited to Gemma family probably due to availability of SAEs?

It would be nice to see similar analysis for different layers of the models in appendix to see generalizability.

**Questions:**

Can experiments matching function and content tokens by frequency but showing different patterns be provided? This would require following the linguistic definition of function versus content words.

---

> ### Author Response · Authors · 2025-11-20
> **Rebuttal by Authors Part1**
>
> We sincerely thank you for your time and effort in reviewing our work. We greatly appreciate your valuable comments and insightful feedback. In our response, we will address each question individually, quoting them and providing our answer accordingly. Please let use know if our responses address your concerns.
>
> ---
> > **Q1: The core definition conflates frequency with linguistic function, making it impossible to determine whether or not effects arise from frequency effects. No experiments match tokens by frequency or justify the arbitrary 40% threshold. Both the choice of 40% threshold and 122 tokens seems arbitrary, and probably contains tokens that would be classified as content tokens in linguistics but are included in their list.**
>
> A1: Thanks for raising this point. We would like to provide several clarifications:
> - How we partition tokens. We divide tokens into function and content based solely on their frequencies in the pre-training corpus (SlimPajama-627B). Language models operate only on token IDs and do not have access to linguistic properties. Tokens with similar distributions in the training data play similar roles from the model's perspective. Therefore, frequency serves as an intrinsic and model-aligned criterion for identifying function tokens.
> - Linguistic categories are not used to define our function and content tokens. We borrow the labels "function" and "content" only as convenient terminology. Our partition is not based on any linguistic properties. As discussed above, linguistic categories are not meaningful to the model, a linguistically defined partition would not be appropriate for our analysis.
> - Why we choose the top 40%. The 40% cutoff is chosen to balance coverage and purity. If we extend the threshold, for example to 50%, admits 300 additional token types, many of which have extremely low document coverage (<10%). These tokens appear frequently only in a small subset of documents in the training corpus, reflecting the "bursty" phenomenon shown in Figure 3(b). Including them would dilute the set and undermine its interpretability as a group of broadly distributed, high-frequency tokens. To preserve a cleaner function token set, we use the 40% threshold.
> - Whether the 122 tokens may contain content word in linguistic. The full list of 122 function tokens is provided in Appendix D for readers who are interested. However, we would like to emphasize that what matters for the model is the distributional pattern in the training corpus, nor their linguistic categories. Our analysis therefore centers on frequency patterns, not linguistic labels.
>
> ---
> > **Q2: Only three features are analyzed in detail in section 3.2 with no statistics across all features.**
>
> A2: Thanks for your feedback. Fully interpreting each feature extracted by an SAE requires extensive manual work, which is impractical. As a result, current interpretability research typically focuses on a limited features. For example, [1] investigates personality control in LLMs using only a few features, such as eval, sycophancy and hallucination. [2] introduces contrastive activation addition and perform experiments on seven features including hallucination, sycophancy, refusal, etc.
> In addition to our case studies, we also provide several statistical analyses, including (1) a token-feature bipartite graph showing that a small number of function tokens activate the majority of features; (2) the growth of learned features through pre-training; (3) pre-training loss curves for different token groups, demonstrating that function->content drives parameter updates in LLMs.
>
> [1] Persona Vectors: Monitoring and Controlling Character Traits in Language Models.
>
> [2] Steering Llama 2 via Contrastive Activation Addition.
>
> ---
> > **Q3: The studies are limited to Gemma family probably due to availability of SAEs?**
>
> A3: Yes. The Gemma family has open-sourced the corresponding SAE models.

---

> ### Author Response · Authors · 2025-11-20
> **Rebuttal by Authors Part2**
>
> > **Q4: It would be nice to see similar analysis for different layers of the models in appendix to see generalizability.**
>
> A4: Thanks for your suggestion. For the token-feature bipartite graph analysis, we selected three representative layers, layer 9 (shallow), layer 20 (middle) and layer 31 (deep). The corresponding results are shown in Figure 5.
> For the case studies, to identify the most informative features for each target trait, we first determine the most informative layer (see more details in Appendix B). Therefore, this experiment is not performed under a fixed-layer setting but rather adapts to the layer provides the strongest signal.
> If there are additional layer-wise analyses you would like to see, please let us know. We would be happy to incorporate them.
>
> ---
> > **Q5: Can experiments matching function and content tokens by frequency but showing different patterns be provided? This would require following the linguistic definition of function versus content words.**
>
> A5: Thanks for your comment. In our analysis, function and content tokens are partitioned based on frequency, so differences in their frequency naturally arise. LLM learns statistical patterns and do not distinguish between linguistically-defined function words and content words during training.

---

### Official Review · Reviewer_Qfmc · 2025-11-01

**Soundness:** 2
**Presentation:** 3
**Contribution:** 1
**Rating:** 2
**Confidence:** 5

**Summary:**

The goal of this paper is to elucidate how memory retrieval and consolidation emerge in large language models. The authors propose the *function token hypothesis*, which posits that during inference, function tokens activate the most predictive features from the context, while during pretraining, predicting content tokens conditioned on function tokens drives the model’s optimization. Here, *function tokens* refer to the most frequent tokens in the vocabulary.

To demonstrate the effectiveness of function tokens, the authors show that these tokens consistently activate the sparsest features in a sparse autoencoder. They further find that the features associated with function tokens can be leveraged for model steering. Finally, by training transformers from scratch, they observe that early stages of learning emphasize the prediction of function tokens, suggesting that such tokens play a key role in shaping the model’s representations.

**Strengths:**

- The paper is well-written and easy to follow.
- I find the authors’ experimental design innovative, as they employ multiple approaches and consider diverse forms of evidence (e.g., SAE features, loss curves).

**Weaknesses:**

- Ultimately, many results in this paper does not provide enough additional insight to the community. For example, in section 4, the authors claim that early training prioritizes predicting function tokens and subsequently, the optimization process becomes dominated by learning to predict content tokens. This seems obvious because function tokens are chosen to be the most frequent tokens.
- Experiments are limited. And I think some claims are not well supported by the experimental evidence. See question section for more.

**Questions:**

- On line 226, you mention that “A token is linked to a feature if it activates the feature in a context”. What exactly does it mean for a feature to be activate? When the feature is nonzero?
- The paper claim that “steering only the activations on the final function token in the prompt (‘\n’) changes the response”. But is it possible that it is due to it being the last position? Overall, I found evidence around this to be extremely weak. The paper only presents case studies without disapproving any alternative hypothesis.
- Also, is the claim that function token just combine features? It is also possible since function tokens activate most tokens. Their features are just mostly noise. So every feature got activated and might not be the effect of any composition.
- Also, in your Chinese example, it seems like the word “Chinese” help steer the answer more than any function tokens because it provides the “speak Chinese features”. Such ideas have been explored in previous papers [1].
- In figure 9, I don’t understand how you can claim that function→content drives the optimization. All curves have the same trend. In fact, green line decreases the least, which could suggest that the model is not learning this well at all.

[1] Jiang, Y., Rajendran, G., Ravikumar, P., & Aragam, B. (2024). Do llms dream of elephants (when told not to)? latent concept association and associative memory in transformers. *Advances in Neural Information Processing Systems*, *37*, 67712-67757.

---

> ### Author Response · Authors · 2025-11-20
> **Rebuttal by Authors Part1**
>
> We sincerely thank you for your effort in reviewing our paper. We greatly appreciate your valuable comments and feedbacks. In our response, we will address each question individually, quoting them and providing our answer accordingly. Please let us know if our responses address your concerns.
>
> ---
> > **Q1: Ultimately, many results in this paper does not provide enough additional insight to the community. For example, in section 4, the authors claim that early training prioritizes predicting function tokens and subsequently, the optimization process becomes dominated by learning to predict content tokens. This seems obvious because function tokens are chosen to be the most frequent tokens.**
>
> A1: We agree that function token prediction is learned faster and more easily since their high frequency in training corpus. In Section 4, we position this as an empirical observation. More importantly, the primary contributions of our work go significantly beyond this phenomenon. Our main contributions are as follow:
> - During inference, function tokens are responsible for activating the most predictive features from the context to govern the next-token prediction. To the best of our knowledge, we are the first to: (1) identify that this capability originates from function tokens, not from any linguistic or semantically special token types, but from their distributional properties in the training corpus; (2) perform systematical experiments on a large-scale token-feature bipartite graph, demonstrating that a relatively small set of function tokens can access the majority of model features; (3) provide detailed case study evidence demonstrating that function tokens can reliably reactivate the most predictive feature from the context, thereby steering next-token prediction.
> - We show that feature growth during pre-training is driven by the prediction of content tokens that follow function tokens. Specifically: (1) we are the first, to our knowledge, to track feature growth throughout pre-training; (2) propose tracking pre-training loss across four token categories, demonstrating function->content drives the optimization.
> - Together, we propose the Function Token Hypothesis for explaining LLM mechanisms of memory retrieval and consolidation.
>
> ---
> > **Q2: On line 226, you mention that “A token is linked to a feature if it activates the feature in a context”. What exactly does it mean for a feature to be activate? When the feature is nonzero?**
>
> A2: We use Sparse Autoencoder (SAE) to decompose the layer activations into features. As shown in Figure 4, in an SAE, each input activation $x\in\mathbb{R}^d$ is encoded into a vector $z\in\mathbb{R}^n$, where each dimension of $z$ corresponds to one learned feature. Here, $d$ is the dimension of the original layer activation and $n$ is the number of features.  The encoding is given by Equation (6):
> $$z=\text{JumpReLU}(W_{\text{enc}}x + b_{\text{enc}})$$
> A feature is considered activated when its corresponding activation value in $z$ is greater than zero. In other words, feature $i$ is activated if $z_i>0$.
>
> ---
> > **Q3: The paper claim that “steering only the activations on the final function token in the prompt (‘\n’) changes the response”. But is it possible that it is due to it being the last position? Overall, I found evidence around this to be extremely weak. The paper only presents case studies without disapproving any alternative hypothesis.**
>
> A3: Thanks for raising the point. To distinguish the effect of the function token itself from the last position, we conduct additional experiments in which we steer activations on other function tokens that are not the final position. The results are shown in Figure 13 in Appendix E in the revised paper. The results show that steering activations on tokens such as 'in', 'of' and 'the' consistently alters the model's output.
> For example, when we steer the token 'in' within the response by manually activating the 'Speak Chinese' and 'UK' features, the model's answer to 'Where is Mount Fuji?' shifts from 'Mount Fuji is located in Japan.' to 'Mount Fuji is located in英国'. This demonstrates that the effect is not restricted to the last-position token and that steering other function tokens can influence the generated text.

---

> ### Author Response · Authors · 2025-11-20
> **Rebuttal by Authors Part2**
>
> > **Q4: Also, is the claim that function token just combine features? It is also possible since function tokens activate most tokens. Their features are just mostly noise. So every feature got activated and might not be the effect of any composition.**
>
> A4: Thanks for your questions. We would like to provide the following clarifications:
> - The Function Token Hypothesis states that "during inference, function tokens reactivate the most predictive features from the context." That means that function tokens selectively reactivate the most predictive features from the context; they are not simply aggregating or combining all features.
> - Function tokens do not activate most features in a given context. Under the superposition phenomenon, each activation vector can be approximated as a sparse linear combinations of features [1]. This is why SAE work: they enforce sparsity. In practice, a standard SAE has an average L0 closest to 100, that means the average activated features of each token is around 100 [2]. Thus, the claim that "function tokens activate most features and their features are mostly noise" is unsupported. Both function and content tokens activate only a small, comparable number of features in a given context.
> - The claim in our paper: a small number of function token can activate most features, refers to the aggregate token-feature bipartite graph. In that graph, a token is connected to a feature if it activates that feature in any context.  The resulting token-degree distribution shows that function tokens have significantly higher degrees than content tokens, revealing an intrinsic difference between the two categories.
> - Concrete example for clarity: In Figure 6(a), 'the' activates the 'speak Chinese' and 'Russia' features. In Figure 6(b), the same token 'the' in a different context activates 'speak Chinese' and 'UK' features, but not the 'Russia' feature. This demonstrate the central point: function token reactivate the most predictive features from the context, and these features differ depending on that context.
>
> [1] Sparse Autoencoders Find Highly Interpretable Features in Language Models, ICLR 2023.
>
> [2]  https://huggingface.co/google/gemma-scope-2b-pt-res
>
> ---
> > **Q5: Also, in your Chinese example, it seems like the word “Chinese” help steer the answer more than any function tokens because it provides the “speak Chinese features”. Such ideas have been explored in previous papers [1].**
>
> A5: Thanks for your comments. We agree that 'Chinese' activates the 'speak Chinese' feature. However, we emphasize that function token plays a central role in governing next token prediction. For example, as shown in Figure 7, only steering the feature activation in the function token is sufficient to directly change the model's next token prediction. This supports our function token hypothesis: function token can reactivate the most predictive features from the context to direct the next-token prediction.
>
> ---
> > **Q6: In figure 9, I don’t understand how you can claim that function→content drives the optimization. All curves have the same trend. In fact, green line decreases the least, which could suggest that the model is not learning this well at all.**
>
> A6: Thanks for raising this point. Although all four curves in Figure 9 show a declining trend, their relative position and convergence behaviors differ in a meaningful way. The function->content has the highest loss in both the 1.5B and 8B models, making it the hardest prediction task. Because this group remains the dominant source of error, the overall optimization is effectively driven by it. This also pushes function tokens to develop the capability to reactivate predictive features from context. In addition, scaling from 1.5B to 8B parameters yields small loss reduction for predicting the next function token, but much larger loss reductions for predicting the next content token. This indicates that scaling model size primarily enhance the content token prediction.

---

> ### Author Response · Authors · 2025-11-26
>
> Thank you for taking the time to review our manuscript. We sincerely appreciate your insightful feedback. As the deadline for the discussion phase approaches, we would like to kindly remind you of our rebuttal, in which we diligently addressed each concern you raised in your feedback.
> If you have any further questions or concerns, we would be happy and eager to address them promptly in the discussion period.
> Thanks again for your valuable feedback.

---

### Meta-Review · Area_Chair_cSK3 · 2025-12-24

**Summary:**

The authors introduce a notion of "function tokens", which corresponding roughly 40% of most frequent tokens. They then hypothesize how these function tokens affect pre-training and inference. In particular, the proposed hypothesis is that these function tokens "activate the most predictive features from context" during inference, and during pre-training they "encourage" memory consolidation. The authors present some experiments to support their hypothesis. For example, they studied how loss differs when the current/next token pairs are different combinations of function/content tokens; the authors performed "causal interventions" to demonstrate that function tokens actively retrieve specific information rather than just passing it along by inserting these function tokens.

**Reviewer Concerns:**

Reviewers raised many concerns. For example, they raised a valid concern that the experiments steering experiments have a confounding variable: token position vs token type. The authors only partially addressed this. The reviewers wanted to see a very specific control: steering a content token in the immediate position (to see if it fails) or steering a distant function token (to see if it succeeds), but this was not done.
The reviewers also noted that the function token definition via thresholding is somewhat arbitrary. The authors proposed some reasoning behind it, but nothing scientifically rigorous. In my opinion, this concern is very valid and unfortunately not addressed.
Somewhat related to the above,  a couple of reviewers argued that the results might be artifacts of language statistics rather than a specialized model mechanism. The authors failed to provide comparative evidence.
The authors argued that some of the concerns around feature activation were due to misundertandings of the key concepts that exist in the literature.

**Reviewer Scores:**

For all reviewers, the key concerns were only partially addressed, as noted above. Therefore, my best guess is that most of them would have recommended a rejection or have given a borderline score.

---

### Decision · Program_Chairs · 2026-01-26

Reject